# Not All Invariants Are Equal: Curating Training Data to Accelerate Program Verification with SLMs

Ido Pinto [1]  Yizhak Yisrael Elboher [1]  Haoze Wu [2 3]  Nina Narodytska [3]  Guy Katz [1]

## Abstract

The synthesis of inductive loop invariants remains a critical bottleneck in automated program verification. While Large Language Models (LLMs) show promise in mitigating this issue, they often fail on complex programs, producing invariants that are invalid or computationally ineffective. Although fine-tuning is a natural strategy to address these limitations, obtaining high-quality training data remains an open challenge. We first formalize the properties required for a high-quality training invariant, and then present WONDA, a rigorous data curation pipeline that extracts such invariants from raw verifier output via AST-based normalization followed by LLM-driven semantic rewriting and augmentation with provable quality guarantees. Fine-tuning Small Language Models (SLMs) on WONDA-curated data yields consistent gains across the Qwen3, Llama-3.1, and Mistral families: the 4B and 8B Qwen3 models nearly double invariant correctness and double speedup rates, while Llama-3.1-8B triples both. On the challenging InvBench suite, the same 4B model outperforms an off-the-shelf model $20\times$ its size and matches the end-to-end verification time of GPT-OSS-120B, while a 14B Qwen3 model matches that of the frontier model GPT-5.2, all without test-time compute overhead. Our code is publicly available on GitHub.

## 1. Introduction

Automated program verification is a powerful technique for ensuring the reliability of critical software infrastructure. At the core of deductive verification lies the challenge of loop invariant synthesis: identifying a logical property that is preserved across every iteration of a loop and is strong enough to prove the program's correctness. Despite decades of research into symbolic methods such as Craig interpolation (McMillan, 2003) and constraint solving (Colón et al., 2003; Gupta & Rybalchenko, 2009), finding inductive invariants remains a challenging problem, and a practical bottleneck for modern verifiers.

The advent of Large Language Models (LLMs) has introduced a new paradigm: *generate-and-verify*. Models such as GPT-4 (OpenAI, 2023) and Claude (Anthropic, 2024) have demonstrated an ability to hypothesize loop invariants (Pei et al., 2023). Recent tools such as Loopy (Kamath et al., 2024), Lemur (Wu et al., 2024b) and ACInv (Liu et al., 2025) leverage this capability, wrapping LLMs in iterative refinement strategies that filter hallucinations using SMT solvers.

Despite this progress, a significant gap remains. As observed by Wei et al. (2025), while LLM-based verifiers represent a promising direction, they do not yet offer a significant advantage over state-of-the-art symbolic tools, such as UAutomizer, which do not leverage LLMs.

Given these limitations, the natural question arises: can we *train* an LLM that specializes in the task of invariant generation? Such specialization is standard practice in many domains and has proven successful. Indeed, within the program verification literature, this specific problem has received increasing attention (Pei et al., 2023; Wei et al., 2025). However, the success of these training efforts has only been partial. Recent work on fine-tuning invariant generation models reports improvements on easy verification problems, yet achieving significant speedups on hard benchmarks remains elusive (Wei et al., 2025). We argue that data quality, rather than model scale alone, is the key bottleneck for current neural invariant synthesizers. Existing datasets (Wei et al., 2025) rely on raw outputs from symbolic tools (e.g., UAutomizer (Heizmann et al., 2013)), which suffer from two limitations:

1. *Low Pedagogical Value:* Solver-generated invariants are often technically correct but structurally obfuscated (see Figure 1), and each output reflects only one valid

[1]Hebrew University of Jerusalem, Israel [2]Amherst College, USA [3]VMware Research by Broadcom, USA. Correspondence to: Ido Pinto <ido.pinto@mail.huji.ac.il>, Guy Katz <g.katz@mail.huji.ac.il>.

*Proceedings of the 43rd International Conference on Machine Learning*, Seoul, South Korea. PMLR 306, 2026. Copyright 2026 by the author(s).

choice among many for the same program. As a result, models may learn verifier-specific artifacts, rather than the underlying program logic or the broader space of useful invariants.

2. *Dependence on Solver Outputs:* When training relies heavily on solver-generated data, the model may inherit the solver's biases, making it harder to improve beyond the current symbolic state-of-the-art.

To mitigate these limitations, we introduce WONDA, a rigorous data curation pipeline. Instead of directly training on verifier-generated invariants, WONDA explicitly optimizes for learnability. We employ Abstract Syntax Tree (AST) normalization to clean the structural representation of invariants. We further implement an LLM-driven semantic rewriting and augmentation engine that transforms obscure machine-generated invariants into concise, interpretable forms, simplifying their logical semantics while maintaining soundness. Augmentation generates multiple candidate rewrites per input, so the model is not limited to the single invariant the solver originally produced. To ensure soundness, we use a formal verification tool to validate invariants after any transformation step that may not preserve correctness.

**Our Contributions.**

1. We demonstrate that directly fine-tuning invariants generated by symbolic solvers does not reliably improve model performance and can even degrade it.

2. We propose WONDA, a rigorous data curation framework that combines AST normalization with LLM-driven semantic rewriting and augmentation, transforming raw, obfuscated symbolic outputs into high-quality training signals while preserving soundness.

3. We present a thorough experimental evaluation across the Qwen3, Llama-3.1, and Mistral model families, showing that WONDA-curated data lets small fine-tuned models match much larger and frontier models on end-to-end verification time, without test-time compute overhead.

## 2. Related Work

**Traditional Invariant Synthesis.** Invariant synthesis has been extensively studied through various formalisms. Abstract Interpretation (Cousot & Cousot, 1977) gave rise to techniques for discovering specific classes of invariants, such as affine relationships (Karr, 1976) and linear restraints (Cousot & Halbwachs, 1978) among variables. Model Checking (Clarke et al., 2018) and Predicate Abstraction (Flanagan & Qadeer, 2002; Lahiri & Bryant, 2007) infer

```
((((((((((((((((((( 1 <= weight1 ) && (((( long long
) weight1 + weight2 ) + 1 ) <= max_threshold )) && (
weight2 <= 10 )) && ( 20 <= (( __int128 ) (( long long )
weight1 * 14 ) + cumulative_weight )))) && ( weight1 <=
10 ) && ( count <= 4 )) && ( 1 <= weight2 )) && ( long
long ) weight1 + weight2 <= cumulative_weight )) || ...
|| ((((((((( 1 <= weight1 ) && (((( long long ) weight1 +
weight2 ) + 1 ) <= max_threshold )) && ( count <= 5 ) &&
( weight2 <= 10 )) && ( weight1 <= 10 )) && ( 20 <= ((
__int128 ) cumulative_weight + (( long long ) weight1 *
13 )))) && ( 1 <= weight2 )) && (( long long ) weight1 +
weight2 <= cumulative_weight )))
```

3084 chars, 75 conjuncts, 11 disjuncts

*Figure 1.* Example of a raw verbose invariant generated by UAutomizer. WONDA transforms such outputs into more compact, learnable forms.

invariants by refining abstract states based on a set of predicates, while Craig Interpolation (McMillan, 2003) generates loop invariants from proofs of unsatisfiability in bounded model checking traces. Tools like UAutomizer (Heizmann et al., 2013) and Eldarica (Hojjat & Rümmer, 2018) build on these approaches. A separate line of work casts invariant generation as a constraint-satisfaction problem solved with off-the-shelf solvers (Colón et al., 2003; Gupta & Rybalchenko, 2009; Fedyukovich & Bodík, 2018). Finally, dynamic analysis tools like Daikon (Ernst et al., 2007) infer likely invariants by observing program execution traces; while efficient, dynamic methods are unsound, as they can only guarantee correctness for the observed executions.

**Learning-based Invariant Synthesis.** Early techniques for data-driven invariant synthesis relied on decision trees (Garg et al., 2016), algebraic inference (Sharma et al., 2013), and Horn-ICE learning (Ezudheen et al., 2018). Pei et al. (2023) explored fine-tuning LLMs for invariant prediction, training on Daikon (Ernst et al., 2007) outputs. With the rise of code-capable LLMs, more recent frameworks employ iterative *generate-and-verify* loops, using symbolic solvers to filter or refine LLM proposals: Loopy (Kamath et al., 2024) applies Houdini-based filtering of LLM-generated candidates, LaM4Inv (Wu et al., 2024a) iteratively queries the LLM and uses BMC to filter and reassemble candidate predicates across rounds, LEMUR (Wu et al., 2024b) introduces backtracking to repair invalid invariants. Related directions include contrastive ranking (Chakraborty et al., 2023), C++ class invariants (Sun et al., 2025), and complex loop structures (Liu et al., 2025).

**Fine-tuning for Invariant Synthesis.** Fine-tuning LLMs to propose a single invariant, subsequently verified by a symbolic solver, has received increasing attention. Unlike Pei et al. (2023), who train on Daikon outputs without formal verification of the generated invariants, Wei et al. (2025) introduce a one-shot setting using UAutomizer-generated training data, which is formally correct but, as we show, structurally noisy (see Figure 1). We adopt the same evalua-

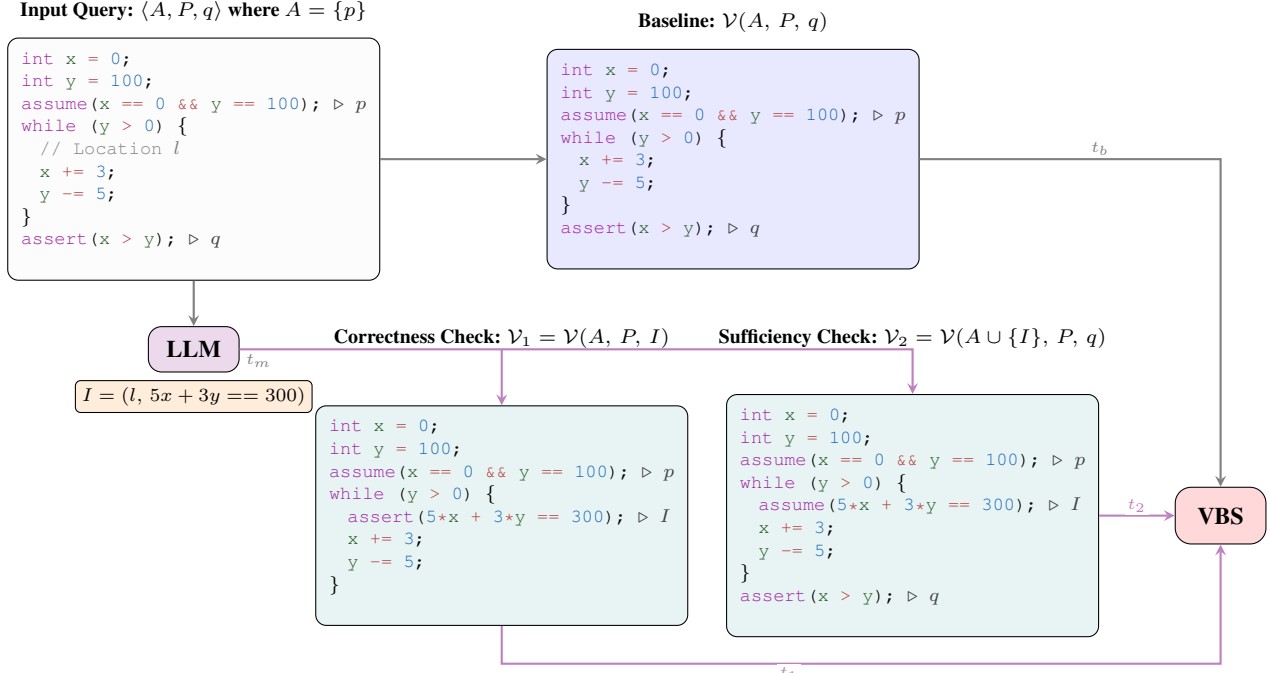

*Figure 2.* Illustration of the VBS metric. Given a query $\langle A, P, q \rangle$, the LLM proposes invariant $I = (l, \varphi)$ with inference latency $t_m$. $\mathcal{V}_1$ and $\mathcal{V}_2$ run in parallel alongside the baseline $\mathcal{V}(A, P, q)$, with $t_v = \max(t_1, t_2)$. VBS selects $\min(t_v, t_b)$ if $I$ is correct and the decomposed checks yield a conclusive result (per Table 1), and falls back to $t_b$ otherwise. For the end-to-end metric $\text{VBP}_{\text{E2E}}$, $t_m$ is added to $t_v$ for sufficient instances.

tion framework and focus on improving training data quality and diversity, producing more compact and generalizable invariants. The one-shot and iterative directions are complementary: while iterative tools such as Lemur (Wu et al., 2024b) and Loopy (Kamath et al., 2024) refine invariants over multiple LLM calls, a stronger one-shot generator provides better initialization for such loops.

# 3. Preliminaries

We ground our approach in Hoare logic (Hoare, 1969), a seminal framework for reasoning about program correctness. A Hoare triple $\{p\} S \{q\}$ asserts that if precondition $p$ holds before executing statement $S$, then postcondition $q$ holds afterward. While sequential statements are straightforward to compose, loops pose a fundamental challenge: they may execute for an unbounded number of iterations, requiring an auxiliary predicate, a *loop invariant*, to enable finite reasoning.

For a loop `while B do S` with precondition $p$ and postcondition $q$, a valid invariant $I$ must satisfy: (i) *Initiation:* $p \Rightarrow I$, meaning the invariant holds upon loop entry; (ii) *Consecution:* $\{I \land B\} S \{I\}$, meaning the invariant is preserved by each iteration; and (iii) *Sufficiency:* $I \land \neg B \Rightarrow q$, meaning that upon termination, the invariant implies the postcondition. A predicate satisfying both

initiation and consecution is termed an *inductive invariant*. Throughout this paper, we refer to this property as *correctness* and use the terms interchangeably; if the predicate additionally satisfies sufficiency, it constitutes a formal proof of the postcondition $q$.

**Running Example.** We illustrate these concepts using the program shown in Figure 2 (top-left block). The program initializes $x = 0$ and $y = 100$ (the precondition $p$), then repeatedly increments $x$ by 3 and decrements $y$ by 5 until $y \leq 0$. The verification goal is to prove the postcondition $q \equiv x > y$. We claim that the candidate invariant $I \equiv 5x + 3y = 300$ satisfies these criteria. It is *inductive* because it holds initially $(5(0) + 3(100) = 300)$ and is preserved by the loop body updates: assuming $I$ holds, the new state satisfies $5(x + 3) + 3(y - 5) = 5x + 15 + 3y - 15 = 5x + 3y = 300$. Furthermore, the invariant is *sufficient*: upon loop termination, $\neg B \equiv y \leq 0$ combined with $I$ implies $5x + 3y = 300 \land y \leq 0$. Substituting, $5x = 300 - 3y \geq 300$ (since $-3y \geq 0$), so $x \geq 60$. As $x \geq 60 > 0 \geq y$, the postcondition $q \equiv x > y$ follows.

## 3.1. Program Verification

Let $P$ denote a program, let $\text{Loc}(P)$ denote its set of program locations, and let $\mathcal{L}(P) \subseteq \text{Loc}(P)$ denote its loop entry locations.

**Definition 3.1** (Property). A *property* is a pair $(l, \varphi)$, where $l \in \text{Loc}(P)$ and $\varphi$ is a predicate over the variables of program $P$. For a given execution of $P$, a property *holds*, or is *satisfied*, if $\varphi$ holds, whenever $P$ reaches line $l$.

**Definition 3.2** (Verification Query). A *verification query* is a triple $\langle A, P, q \rangle$ where $P$ is a program, $q$ is a property (the postcondition), and $A$ is a set of properties representing preconditions.

**Definition 3.3** (Verification Query Validity). A verification query $\langle A, P, q \rangle$ is *valid* if every execution of $P$ that satisfies the (precondition) properties in $A$ also satisfies the postcondition $q$.

**Definition 3.4** (Verification Oracle). A *verification oracle* $\mathcal{V}$ takes a verification query $\langle A, P, r \rangle$ and returns:

$$\mathcal{V}(A, P, r) \in \{\text{TRUE}, \text{FALSE}, \text{UNKNOWN}\}$$

where TRUE indicates the query is valid, FALSE indicates a counterexample exists, and UNKNOWN indicates the oracle could not determine the result (e.g., due to timeout or inherent incompleteness). Here, we consider only *sound* oracles: if $\mathcal{V}(A, P, r) = \text{TRUE}$, then the query is valid; if $\mathcal{V}(A, P, r) = \text{FALSE}$, then a counterexample exists.

Verification queries use `assume(`$\varphi$`)` and `assert(`$\varphi$`)` statements. `assume(`$\varphi$`)` at line $l$ restricts traces to those satisfying $\varphi$ (equivalent to `if (`$\neg\varphi$`) halt`), while `assert(`$\varphi$`)` jumps to ERROR if violated (equivalent to `if (`$\neg\varphi$`) goto ERROR`). For query $\langle A, P, q \rangle$, we annotate $P$ with `assume(`$\varphi$`)` for each $(l, \varphi) \in A$ and `assert(`$\varphi$`)` where $q = (l, \varphi)$. $P$ is safe with respect to the verification query if and only if ERROR is unreachable.

### 3.2. Program Verification using Invariants

Given a verification query $\langle A, P, q \rangle$, the task can be addressed through a direct invocation of the verifier: $\mathcal{V}(A, P, q)$. Alternatively, we can break the problem into sub-problems, by introducing a candidate invariant property $I = (l, \varphi)$, where $l \in \mathcal{L}(P)$. This approach splits the verification task into two distinct queries:

1. *Correctness Check* ($\mathcal{V}_1$): Verify that the property $I$ is correct (inductive) within the program $P$ given $A$: $\mathcal{V}_1 := \mathcal{V}(A, P, I)$

2. *Sufficiency Check* ($\mathcal{V}_2$): Verify that the target postcondition $q$ holds, assuming the correctness of the candidate invariant $I$: $\mathcal{V}_2 := \mathcal{V}(A \cup \{I\}, P, q)$

We denote by $t_b$ the wall-clock time of direct verification $\mathcal{V}(A, P, q)$, and by $t_v = \max(t_1, t_2)$ the parallel execution time of the correctness and sufficiency checks, where $t_1$ and $t_2$ are their respective solving times.

The final outcome is determined by Table 1. This procedure is sound; conclusive outcomes (TRUE or FALSE) are guaranteed to be correct (Wei et al., 2025). This approach augments the verification problem with new knowledge that can be useful to prove the desired property. The correctness and sufficiency checks are independent and can therefore be performed in parallel, and in the case where both queries are simpler than the original query, the wall-clock verification time is reduced.

*Table 1.* Decision procedure for verification queries using a candidate invariant. Note that a FALSE outcome in the sufficiency check implies the original query is invalid regardless of the invariant's correctness.

| $\mathcal{V}_1$ (Correctness) | $\mathcal{V}_2$ (Sufficiency) | Outcome |
|---|---|---|
| TRUE | TRUE | TRUE |
| * | FALSE | FALSE |
| otherwise | | UNKNOWN |

An illustration of the approach appears in Figure 2. Instead of directly verifying the query (Baseline block, top-right), we can use the invariant $I = (l, 5x + 3y = 300)$ and verify its correctness (Correctness Check block, bottom-left) and sufficiency (Sufficiency Check block, bottom-right). When the verifier returns TRUE for both of these queries, we can immediately deduce the safety of the program.

**Generating Invariants.** The effectiveness of the aforementioned techniques is conditional upon our ability to generate useful invariant candidates, i.e., candidates that will allow us to reach a TRUE/FALSE outcome, as per Table 1. We define the invariant generation task, which is the primary focus of this work, as follows: given a verification query $\langle A, P, q \rangle$, a verification oracle $\mathcal{V}$, and a designated loop entry $l \in \mathcal{L}(P)$, the objective is to synthesize a predicate $\varphi$ such that the resulting property $I = (l, \varphi)$ satisfies the correctness and sufficiency checks; i.e., $\mathcal{V}$ returns TRUE for the corresponding correctness and sufficiency queries. We also regard a correct candidate whose sufficiency check returns FALSE as desirable, since this witnesses a genuine bug in $P$.

## 4. Methodology

The main component of the *generate-and-verify* paradigm is the LLM that performs invariant synthesis. In practice, off-the-shelf LLMs are not very effective at this task and need to be fine-tuned for the invariant generation domain. Somewhat counterintuitively, fine-tuning on data produced by modern verification tools does not show a performance boost on challenging problems (Wei et al., 2025). We argue that fine-tuning can in fact be quite useful, provided the training data is of high quality. To this end, we investigate two research questions: (a) how to define high-quality training data in this context; and (b) how to produce it.

**$V_0$: Raw Verifier Output**

```
((((((((((((((((((( 36 <= y )
&& ( 36 <= x ))
 || (( 21 <= y ) && ( 45 <= x )))
 || (( 21 <= x ) && ( 65 <= y )))
 || (( 42 <= x ) && ( 26 <= y )))
 || (( 12 <= x ) && ( 80 <= y )))
 || (( 11 <= y ) && ( 51 <= x )))
      .
      .
      .
 || (( 16 <= y ) && ( 48 <= x )))
```

603 chars, 21 disjuncts

**$V_1$: Normalized**

```
36 <= y && 36 <= x
|| 21 <= y && 45 <= x
|| 21 <= x && 65 <= y
|| 42 <= x && 26 <= y
|| 12 <= x && 80 <= y
      .
      .
      .
|| 16 <= y && 48 <= x
```

441 chars, 21 disjuncts

**LLM**

*normalize*   *simplify*

**$V_2$: Simplified**

```
5*x + 3*y == 300
```

18 chars, 1 equation

*Figure 3.* WONDA pipeline. The raw verifier output ($V_0$) is normalized ($V_1$), then an LLM simplifies it to a compact closed-form expression ($V_2$): $5x + 3y = 300$, achieving a **2x** verification speedup.

Given a verification query $\langle A, P, q \rangle$, a verification oracle $\mathcal{V}$ and loop locations $\mathcal{L}(P)$, our goal is to produce training samples that pair programs with high-quality loop invariants. We claim that a good invariant $I = (l, \varphi)$ should satisfy the following characteristics:

1. *Non-Degeneracy:* we exclude trivial invariants $\varphi \in \{\text{FALSE}, \text{TRUE}\}$.

2. *Correctness:* invariant $\varphi$ holds at location $l$ on all executions, i.e., $\mathcal{V}(A, P, I) = \text{TRUE}$.

3. *Usefulness:* using $I$ expedites verification times. This entails sufficiency, i.e., $\mathcal{V}_2 = \text{TRUE}$, and that $t_v < t_b$, where $t_v$ and $t_b$ are as defined in Section 3.2.

4. *Compactness:* invariant $I$ has a succinct syntactic form, which we hypothesize facilitates learning and improves generalization.

**Grounding to Verifier-Generated Invariants.** Following Wei et al. (2025), we initialize our curation process using invariants extracted from UAutomizer. When UAutomizer proves a verification query $\langle \{p\}, P, q \rangle$, it emits discovered invariants $\{(l, \varphi_{\text{raw}})\}$ as part of its proof. These invariants satisfy *correctness* by construction. However, they are not guaranteed to be *useful* (they may not provide speedup) or *compact* (they often contain tool-specific artifacts). Figure 1 depicts an example: while correct, the raw invariant is cluttered with type casts and verbose structure, making it a poor training target.

We address these issues with a two-stage pipeline:

1. *Invariant Normalization* (§4.1): AST-based rewriting that removes tautologies, contradictions, minimizes parentheses and strips redundant type casts.

2. *LLM-Based Simplification* (§4.2): A data augmentation step where an LLM generalizes verbose invariants into compact candidates, each verified for correctness and usefulness.

Figure 3 illustrates output produced by our pipeline. Another example appears in Figure 6 (Appendix A).

### 4.1. Invariant Normalization

Raw invariants from automated verifiers are often cluttered with syntactic noise that obscures underlying program logic. This noise typically manifests as: (i) tautological clauses (e.g., `n <= n`); (ii) redundant parenthesization, and (iii) excessive integral type casts (e.g., `__int128` and `long long`) used for bit-precise semantics. In rare cases, verifiers may even produce contradictions within unreachable or dead-code paths, such as `(x > x)`, which can confuse models during training.

We apply a semantic-preserving AST normalization, NOR-MALIZE, which performs a single bottom-up traversal using the rules in Table 2. It also strips the invariants of any unnecessary parentheses, based on standard C operator precedence rules.

*Table 2.* Rewrite rules applied during AST normalization. $e$ represents a numeric variable, $\varphi$ a predicate, $c$ a constant, and $\bowtie \in \{\leq, \geq, =, <, >, \neq\}$ a relational operator.

| Rule | Transformation |
|---|---|
| *Tautology Elimination* | |
| TAUTCONJ | $\varphi \wedge true$ or $true \wedge \varphi \longrightarrow \varphi$ |
| TAUTREFL | $e \bowtie e \longrightarrow true$ for $\bowtie \in \{\leq, \geq, =\}$ |
| TAUTCONST | $c_1 \bowtie c_2 \longrightarrow true$ if $c_1 \bowtie c_2$ holds |
| *Contradiction Propagation* | |
| CONTRACONJ | $\varphi \wedge false$ or $false \wedge \varphi \longrightarrow false$ |
| CONTRADISJ | $\varphi \vee false$ or $false \vee \varphi \longrightarrow \varphi$ |
| CONTRAREFL | $e \bowtie e \longrightarrow false$ for $\bowtie \in \{<, >, \neq\}$ |

**Proposition 4.1** (Normalization Soundness). *For any predicate $\varphi$, the normalized predicate $\varphi' = \text{NORMALIZE}(\varphi)$ is semantically equivalent to the original ($\varphi' \equiv \varphi$).*

*Proof.* The rewrite rules implement standard first-order logic identities applied inductively via bottom-up AST

traversal. The parenthesis minimization follows the C operator precedence while preserving associativity and binding. □

As an additional optimization step, we eliminate integral casts to reduce clutter. While stripping casts can break soundness (e.g., under certain overflow conditions), we prioritize logical clarity for model training and formally verify the final generated invariants against the original program to ensure correctness is maintained.

## 4.2. LLM-Based Invariant Simplification

AST normalization removes syntactic noise, but many invariants remain verbose due to *semantic* complexity: enumerated cases, redundant bounds, or overly strong constraints. These patterns require reasoning beyond local rewrites. To bridge this gap, we employ an LLM as a *simplification function* $f_\theta : \Phi \to \Phi^N$, where $\Phi$ is the set of possible predicates and $N$ is the number of generated candidates. Unlike the rules in Table 2, $f_\theta$ is not guaranteed to be semantics-preserving ($f_\theta(\varphi) \not\equiv \varphi$). Instead, we leverage the LLM's ability to perform *abstraction*, aiming for candidates that are more compact yet still sufficient to prove property $q$. The resulting predicate can be equivalent, stronger, weaker or incomparable with the original predicate; this does not jeopardize soundness, as we later invoke the verifier to ensure the correctness and sufficiency of the rewritten predicates.

**Simplification Procedure.** To guide the LLM toward high-quality invariants, we provide a prompt context $\mathcal{C} = \langle P, \varphi, \mathcal{G} \rangle$ containing the program $P$, the reference invariant $\varphi$, and a set of transformation goals $\mathcal{G}$. The following transformation goals capture our main insights for achieving the desired characteristics of *compactness* and *usefulness*.

*Range generalization.* We observed that a large number of raw invariants have a case-enumeration structure; for example, $\bigvee_{i=1}^{k}(x = i)$ might come from a loop that runs for $k$ steps. We prompt an LLM to simplify such invariants into a more compact range invariant, $1 \leq x \leq k$, which is easier to learn than the enumeration representation.

*Constraint factoring.* Some invariants form Boolean expressions that have unnecessarily complex logical structure, e.g., $(a \wedge b_1) \vee \ldots \vee (a \wedge b_k)$. We encourage an LLM to perform logical simplification, e.g., rewrite this as $a \wedge (b_1 \vee \ldots \vee b_k)$.

*Closed-form discovery.* Often, we would like to generalize the normalized invariant to replace case enumerations with arithmetic relations, where linear expressions are preferred for solver efficiency, though non-linear closed forms are also possible. This gives us compact, reusable invariants that are easier to learn; for example, $\sum_{i=1}^{n} i \to \frac{n(n+1)}{2}$.

*Redundancy removal.* Some expressions are implied by

program semantics and hold throughout the program, e.g., preconditions or variables with fixed values. To simplify them, we can remove these redundant clauses to obtain more compact and easier-to-learn invariants.

Overall, our strategy encourages semantic abstraction over syntactic rewriting; for the full prompt, see Appendix B.2.

**Definition 4.2** (Quality Grade $G$)**.** To measure the utility of a candidate $\varphi'$ with respect to a verification query $\mathcal{Q} = \langle A, P, q \rangle$ and baseline time $t_b$, we define a grading function $G(\varphi', \mathcal{Q}, t_b) \in \{0, 1, 2, 3\}$ based on $\mathcal{V}_1, \mathcal{V}_2$ checks defined in §3.2.

$$G(\varphi', \mathcal{Q}, t_b) = \begin{cases} 0 & \mathcal{V}_1 \neq \text{TRUE} \\ 1 & \mathcal{V}_1 = \text{TRUE} \wedge \mathcal{V}_2 \neq \text{TRUE} \\ 2 & \mathcal{V}_1 = \text{TRUE} \wedge \mathcal{V}_2 = \text{TRUE} \wedge t_v \geq t_b \\ 3 & \mathcal{V}_1 = \text{TRUE} \wedge \mathcal{V}_2 = \text{TRUE} \wedge t_v < t_b \end{cases}$$

where $t_v = \max(t_1, t_2)$ and $t_b$ are as defined in Section 3.2. The full grading procedure is given in Algorithm 1 (Appendix C). Syntactically invalid candidates are discarded before grading. We retain candidates with $G(\varphi', \mathcal{Q}, t_b) \geq 2$ as "golden" training samples.

The simplification procedure is given in Algorithm 2 (Appendix C). It first discards degenerate invariants, i.e., $\varphi_{\text{norm}} \in \{\text{FALSE}, \text{TRUE}\}$. Otherwise, if the invariant is deemed verbose (in our implementation, if $|\varphi_{\text{norm}}| > \eta$ for a threshold $\eta > 0$), an LLM generates $N$ candidates. These candidates are deduplicated by exact match, filtered again for degeneracy, and each is graded using Algorithm 1; we retain those with grade $g \geq 2$ as pairs $(\varphi, g)$. If no candidate qualifies (or if $\varphi_{\text{norm}}$ is not verbose), we instead grade $\varphi_{\text{norm}}$ itself.

**Proposition 4.3** (Pipeline Soundness)**.** *If the pipeline outputs $(l, \varphi^*)$, then $\varphi^*$ is a correct loop invariant sufficient to prove q.*

*Proof.* By construction, $\varphi^*$ is returned only if it passes the formal verifier's correctness check ($\mathcal{V}_1 = \text{TRUE}$) and sufficiency check ($\mathcal{V}_2 = \text{TRUE}$). □

## 5. Experimental Setup

**Training Dataset.** We fine-tune on raw verifier-generated invariants extracted from training programs released by Wei et al. (2025). We run UAutomizer to verify each program and collect loop invariants from its output; we denote this raw collection *V0*. AST-based normalization yields *V1*, and LLM-driven simplification (Kimi K2 Thinking (Team et al., 2025), $N=4$ candidates per verbose invariant, where $\eta = 20$ characters) followed by verifier filtering yields *V2*. Retaining only $g \geq 2$ candidates for fine-tuning yields 7,284 samples, partitioned 80/20 into train and validation. Full pipeline yield and dataset statistics are in Appendix D.

**Evaluation Dataset.** For evaluation, we use InvBench, a benchmark of C verification queries derived from SV-COMP (Beyer & Strejcek, 2025), released by Wei et al. (2025). Starting from 219 programs, we expand multi-loop programs into per-loop instances, yielding 362 total. We partition these into *Easy* ($n = 239$, 66%) and *Hard* ($n = 123$, 34%) using a 15-second UAutomizer baseline threshold, focusing on Hard instances where invariants provide meaningful speedup; 20 Hard instances time out (Appendix E). Each instance in $\mathcal{D} = \{(p_i, P_i, q_i, l_i, t_b^{(i)})\}_{i=1}^n$ comprises a precondition $p_i$, program $P_i$, postcondition $q_i$, loop location $l_i \in \mathcal{L}(P_i)$, and median baseline time $t_b^{(i)}$ over $k = 3$ runs. Models are prompted with a structured JSON format (Appendix B.1); syntactically invalid, ill-formed, or outputs using side-effect operators (e.g., $++, + =, =$) receive verdict UNKNOWN. UAutomizer configuration, and hardware used for evaluation are detailed in Appendix G.

**Models.** We evaluate three open model families. From *Qwen3* (Yang et al., 2025) we use Qwen3-0.6B, Qwen3-4B-Instruct-2507, Qwen3-8B, and Qwen3-14B (referred to as *Qwen3-0.6B/4B/8B/14B*); we also use *Llama-3.1-8B-Instruct* (Grattafiori et al., 2024) and *Mistral-7B-Instruct-v0.3* (Jiang et al., 2023), referred to as *Llama-3.1-8B* and *Mistral-7B*. All models are fully fine-tuned except Qwen3-8B, which uses LoRA (Hu et al., 2022) (Appendix F). As off-the-shelf baselines, we compare against Qwen3-Next-80B-A3B-Instruct (referred to as *Qwen3-80B*), GPT-OSS-120B (Agarwal et al., 2025), and GPT-5.2 (OpenAI, 2025). The Qwen3-0.6B, 8B, and 14B models are trained and evaluated in *non-thinking mode*.

**Decision Procedure.** For valid candidates, we apply the procedure from §3.2, executing two queries *in parallel*:

$$\mathcal{V}_1^{(i)} = \mathcal{V}(\{p_i\}, P_i, (l_i, \hat{\varphi}_i)) \qquad \text{(Correctness Check)}$$
$$\mathcal{V}_2^{(i)} = \mathcal{V}(\{p_i, (l_i, \hat{\varphi}_i)\}, P_i, q_i) \quad \text{(Sufficiency Check)}$$

with wall-clock times $t_1^{(i)}$ and $t_2^{(i)}$, respectively. Since both queries execute in parallel, the verification time is $t_v^{(i)} = \max(t_1^{(i)}, t_2^{(i)})$. The outcome $D_i$ is determined by Table 1.

**Evaluation Metrics.** We define binary indicators for each instance $i$:

1. VALID$(i) = 1$ iff syntactic validation passes;

2. CORRECT$(i) = 1$ iff VALID$(i)$ and $\mathcal{V}_1^{(i)} = $ TRUE;

3. SPEEDUP$(i) = 1$ iff CORRECT$(i)$, $D_i \in \{\text{TRUE}, \text{FALSE}\}$, and $t_v^{(i)} < t_b^{(i)}$.

We report the mean of each indicator across the evaluation set. We also define the speedup factor:

$$S(i) = \begin{cases} t_b^{(i)}/t_v^{(i)} & \text{if CORRECT}(i) \wedge \\ & \qquad D_i \in \{\text{TRUE}, \text{FALSE}\} \\ 1 & \text{otherwise} \end{cases}$$

We report $\bar{S}_{>1}$ as the mean speedup factor among instances with SPEEDUP$(i) = 1$.

**Virtual Best Solver.** The verification procedure described in Section 3.2 is designed to run as part of a portfolio, in parallel to direct verification: the LLM proposes an invariant while the baseline verifier runs concurrently, and whichever finishes first determines the outcome. The *Virtual Best Solver (VBS)* metric captures this by selecting the faster strategy per instance (cf. Figure 2):

$$\text{VBS}(i) = \begin{cases} \min(t_v^{(i)}, t_b^{(i)}) & \text{if CORRECT}(i) \wedge \\ & \qquad D_i \in \{\text{TRUE}, \text{FALSE}\} \\ t_b^{(i)} & \text{otherwise} \end{cases}$$

The *Virtual Best Performance* VBP $= \frac{1}{n} \sum_i \text{VBS}(i)$ measures the average verification time achievable by optimally combining LLM invariants with baseline direct verification.

**Note on Model Latency.** Our primary metrics ($R_{\text{speedup}}$, $\bar{S}_{>1}$, and VBP) exclude inference latency $t_m^{(i)}$ to isolate model capability from deployment factors. For end-to-end cost, we additionally report VBP$_{\text{E2E}}$, which adds $t_m^{(i)}$ to $t_v^{(i)}$ for sufficient instances.

## 6. Results and Analysis

**Benefits of WONDA.** Table 3 depicts our main results on the hard split. Across all open-model scales, V2 curation significantly boosts performance relative to the corresponding base. *Qwen3-4B-V2* more than doubles its base model's speedup rate and nearly doubles its correctness, with VBP$_{\text{E2E}}$ falling from $185.7\,s$ to $165.7\,s$. *Qwen3-8B-V2* shows the same pattern, *Llama-3.1-8B-V2* triples base correctness ($14.9\% \rightarrow 45.5\%$), and *Mistral-7B-V2* more than doubles its speedup rate ($7.3\% \rightarrow 16.0\%$). *Qwen3-14B-V2* achieves the best VBP among our models ($162.1\,s$), matching GPT-5.2 on VBP$_{\text{E2E}}$ ($162.9\,s$ vs. $163.4\,s$). VBP improves by up to 16.0% ($\sim 31\,s$) over direct verification alone across all families. All fine-tuned Qwen3 models at 4B parameters and above, as well as *Llama-3.1-8B-V2*, outperform the off-the-shelf *Qwen3-80B* on both correctness and VBP$_{\text{E2E}}$, with *Qwen3-4B-V2* the most notable given it is $\sim 20\times$ smaller; it also matches GPT-OSS-120B on VBP$_{\text{E2E}}$ ($165.7\,s$ vs. $167.6\,s$). Figure 4 summarizes these comparisons visually.

*Table 3.* Main results on the Hard instances ($n = 123$). Results shown as mean $\pm$ std. across three runs. $R_{\text{valid/correct/speedup}}$: indicator rates (%). $\bar{S}_{>1}$: mean speedup among accelerated instances. VBP: Virtual Best Performance in seconds (verifier-only baseline VBP: 193s). Solved: baseline timeouts (of 20) resolved per run. **Bold**: indicates best per model scale.

| Model | $R_{\text{valid}}$ (%) | $R_{\text{correct}}$ (%) | $R_{\text{speedup}}$ (%) | $\bar{S}_{>1}$ (x) | VBP $\downarrow$ (s) | VBP$_{\text{E2E}}$ $\downarrow$ (s) | Solved |
|---|---|---|---|---|---|---|---|
| GPT-5.2 | 94.0$\pm$1.7 | 72.4$\pm$2.2 | 37.1$\pm$1.2 | 10.7$\pm$0.4 | 155.6$\pm$3.0 | 163.4$\pm$3.0 | 3, 2, 3 |
| GPT-OSS-120B | 92.1$\pm$1.2 | 58.0$\pm$1.2 | 27.4$\pm$2.9 | 7.0$\pm$1.4 | 165.8$\pm$5.6 | 167.6$\pm$5.7 | 3, 2, 1 |
| Qwen3-80B | 97.8$\pm$0.5 | 38.8$\pm$1.7 | 21.4$\pm$1.2 | 9.5$\pm$1.0 | 169.5$\pm$2.1 | 169.7$\pm$2.1 | 4, 3, 3 |
| Qwen3-14B | 96.5$\pm$0.5 | 36.3$\pm$1.9 | 13.6$\pm$2.0 | 7.6$\pm$0.9 | 183.2$\pm$1.2 | 183.6$\pm$1.1 | 1, 2, 1 |
| Qwen3-14B-V2 (Ours) | **100.0$\pm$0.0** | **43.4$\pm$4.9** | **18.4$\pm$4.2** | **16.0$\pm$3.3** | **162.1$\pm$8.3** | **162.9$\pm$8.1** | **4, 4, 2** |
| Qwen3-8B (Base) | 89.4$\pm$7.8 | 23.8$\pm$3.1 | 10.8$\pm$0.5 | 8.5$\pm$5.2 | 181.6$\pm$4.3 | 181.7$\pm$4.2 | 0, 0, 3 |
| Qwen3-8B-V2 (Ours) | **100.0$\pm$0.0** | **42.8$\pm$4.6** | **21.7$\pm$1.7** | **10.7$\pm$2.3** | **166.5$\pm$4.3** | **166.7$\pm$4.3** | **2, 1, 4** |
| Qwen3-4B (Base) | 99.2$\pm$0.0 | 22.8$\pm$2.2 | 11.1$\pm$0.9 | 8.9$\pm$2.5 | 185.6$\pm$2.4 | 185.7$\pm$2.4 | 1, 0, 1 |
| Qwen3-4B-V2 (Ours) | **100.0$\pm$0.0** | **44.4$\pm$2.3** | **24.7$\pm$1.2** | **12.4$\pm$2.2** | **165.5$\pm$3.2** | **165.7$\pm$3.2** | **3, 2, 2** |
| Qwen3-0.6B (Base) | 88.3$\pm$0.5 | **28.5$\pm$2.8** | 12.2$\pm$2.2 | 5.3$\pm$3.3 | 182.9$\pm$5.7 | 183.0$\pm$5.7 | 2, 0, 1 |
| Qwen3-0.6B-V2 (Ours) | **99.7$\pm$0.5** | 27.9$\pm$0.5 | **14.1$\pm$2.5** | **8.5$\pm$3.1** | **174.0$\pm$5.6** | **174.1$\pm$5.6** | 2, 2, 1 |
| Llama3.1-8B (Base) | 96.2$\pm$1.2 | 14.9$\pm$1.7 | 5.4$\pm$2.0 | 3.7$\pm$1.4 | 186.3$\pm$5.1 | 186.4$\pm$5.1 | 1, 0, 2 |
| Llama3.1-8B-V2 (Ours) | **99.7$\pm$0.5** | **45.5$\pm$1.4** | **18.4$\pm$2.5** | **11.3$\pm$2.1** | **168.9$\pm$7.6** | **169.2$\pm$7.5** | **3, 2, 4** |
| Mistral-7B (Base) | 93.8$\pm$0.9 | 18.4$\pm$2.0 | 7.3$\pm$1.4 | 6.7$\pm$3.6 | 186.7$\pm$1.5 | 186.9$\pm$1.5 | 0, 0, 0 |
| Mistral-7B-V2 (Ours) | **99.5$\pm$0.9** | **31.4$\pm$0.5** | **16.0$\pm$0.5** | **17.0$\pm$0.9** | **169.0$\pm$4.3** | **169.4$\pm$4.3** | **2, 1, 4** |

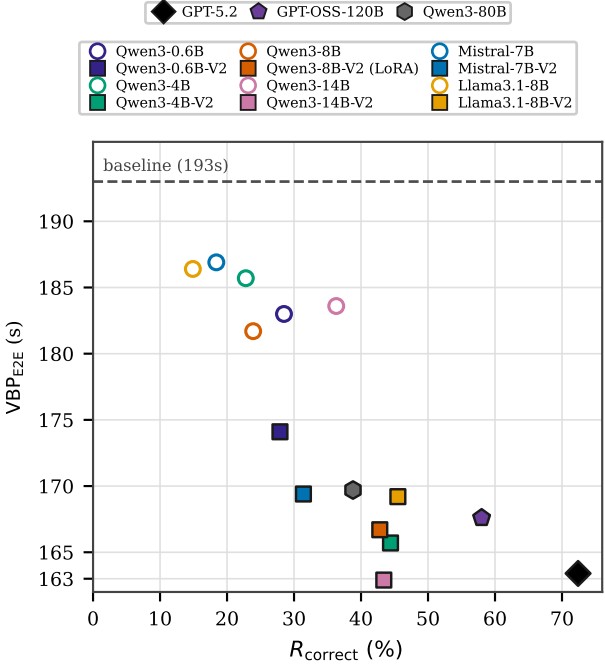

*Figure 4.* **Correctness vs.** VBP$_{\text{E2E}}$ on hard split ($n$=123; mean over three runs). Dashed line: baseline VBP (193 s); better is bottom-right. V2 improves both axes across most of the models; *Qwen3-4/8B-V2* match *GPT-OSS-120B* on VBP$_{\text{E2E}}$, with *Qwen3-14B-V2* matching *GPT-5.2*, despite lower correctness rate.

**Invariant Correctness vs. Verification Speedup.** Figure 4 summarizes the joint improvement in correctness and VBP$_{\text{E2E}}$ across families: in most cases, V2 (squares) improves over base (circles) on both axes. However, a cor-

rect invariant need not accelerate verification; Table 3 separates $R_{\text{correct}}$ from $R_{\text{speedup}}$ and VBP$_{\text{E2E}}$ for this reason. *Qwen3-0.6B* illustrates the decoupling most clearly: base and V2 achieve similar $R_{\text{correct}}$ (28.5% vs. 27.9%), yet V2 yields higher $R_{\text{speedup}}$ (14.1% vs. 12.2%), larger $\bar{S}_{>1}$ (8.5$\times$ vs. 5.3$\times$), and lower VBP$_{\text{E2E}}$ (174.1 $s$ vs. 183.0 $s$). WONDA targets this gap by curating invariants that are not only correct but also practically beneficial to the verifier.

**Timeouts remain a challenge.** Resolving baseline timeouts remains an open problem: even the strongest models clear only a small fraction of the 20 hard-split instances that timed out (Table 3, *Solved*). Still, V2 makes measurable progress. *Qwen3-14B-V2* averages $\sim$3.3 resolved per run, the highest among all models and above GPT-5.2 ($\sim$2.7); *Llama-3.1-8B-V2* also exceeds GPT-5.2 at $\sim$3.0. *Mistral-7B* base stays at 0, while *Mistral-7B-V2* averages $\sim$2.3 (up to 4). The consistent base$\rightarrow$V2 improvement across families suggests that WONDA teaches invariants that help the verifier resolve previously intractable instances.

**Ablation Analysis.** Table 4 traces *V0$\rightarrow$V1$\rightarrow$V2* on Qwen3-4B/8B, Llama-3.1-8B, and Mistral-7B. *V0* fine-tuning on raw verifier-generated invariants often *lowers* $R_{\text{valid}}$ (e.g., Qwen3-4B 99.2% $\rightarrow$ 81.3%; Mistral-7B 93.8% $\rightarrow$ 65.0%); *V1* normalization restores syntax but yields only modest gains. *V2* brings the largest improvements across all families: correctness and speedup rates nearly double on Qwen3-4B and 8B, Llama-3.1-8B-V2 reaches the highest $R_{\text{correct}}$ in the table (45.5%), and Mistral-7B recovers from the V0 drop. Only the full WONDA pipeline consistently delivers both syntactic reliability and meaningful verification speedup. This trend is illustrated concretely in Figure 5 in

*Table 4.* WONDA ablation study on the hard split ($n=123$; mean $\pm$ std. over three runs). *V0*: raw UAutomizer invariants; *V1*: AST-normalized; *V2*: full pipeline. **Bold**: best per model family.

| Model | $R_{valid}$ (%) | $R_{correct}$ (%) | $R_{speedup}$ (%) | $\bar{S}_{>1}$ (x) | VBP $\downarrow$ (s) | VBP$_{E2E}$ $\downarrow$ (s) |
|---|---|---|---|---|---|---|
| Qwen3-8B (Base) | 89.4±7.8 | 23.8±3.1 | 10.8±0.5 | 8.5±5.2 | 181.6±4.3 | 181.7±4.2 |
| Qwen3-8B-V0 | 88.1±3.9 | 29.9±3.4 | 11.5±1.9 | 9.4±2.6 | 180.0±2.7 | 180.6±2.7 |
| Qwen3-8B-V1 | 97.0±0.9 | 30.1±0.8 | 13.0±2.2 | 9.1±1.9 | 175.3±3.2 | 175.5±3.2 |
| Qwen3-8B-V2 (Ours) | **100.0±0.0** | **42.8±4.6** | **21.7±1.7** | **10.7±2.3** | **166.5±4.3** | **166.7±4.3** |
| Qwen3-4B (Base) | 99.2±0.0 | 22.8±2.2 | 11.1±0.9 | 8.9±2.5 | 185.6±2.4 | 185.7±2.4 |
| Qwen3-4B-V0 | 81.3±0.8 | 29.3±3.5 | 13.6±1.9 | 10.1±1.5 | 177.5±1.5 | 177.7±1.5 |
| Qwen3-4B-V1 | 97.6±1.4 | 33.1±2.3 | 12.7±2.3 | 11.4±2.9 | 174.2±4.7 | 174.4±4.7 |
| Qwen3-4B-V2 (Ours) | **100.0±0.0** | **44.4±2.3** | **24.7±1.2** | **12.4±2.2** | **165.5±3.2** | **165.7±3.2** |
| Qwen3-0.6B (Base) | 88.3±0.5 | **28.5±2.8** | 12.2±2.2 | 5.3±3.3 | 182.9±5.7 | 183.0±5.7 |
| Qwen3-0.6B-V0 | 85.9±2.6 | 18.7±0.8 | 8.9±0.8 | 11.7±9.4 | 178.0±2.7 | 178.1±2.7 |
| Qwen3-0.6B-V1 | 97.6±1.4 | 23.3±1.2 | 9.8±2.2 | **15.3±0.9** | 174.4±4.4 | 174.5±4.4 |
| Qwen3-0.6B-V2 (Ours) | **99.7±0.5** | 27.9±0.5 | **14.1±2.5** | 8.5±3.1 | **174.0±5.6** | **174.1±5.6** |
| Llama3.1-8B (Base) | 96.2±1.2 | 14.9±1.7 | 5.4±2.0 | 3.7±1.4 | 186.3±5.1 | 186.4±5.1 |
| Llama3.1-8B-V0 | 88.9±0.5 | 31.2±2.0 | 14.9±0.9 | 9.3±3.1 | 175.0±2.5 | 175.4±2.5 |
| Llama3.1-8B-V1 | 99.7±0.5 | 36.3±5.4 | 14.1±3.7 | **15.3±3.4** | 170.0±3.5 | 170.4±3.5 |
| Llama3.1-8B-V2 (Ours) | 99.7±0.5 | **45.5±1.4** | **18.4±2.5** | 11.3±2.1 | **168.9±7.6** | **169.2±7.5** |
| Mistral-7B (Base) | 93.8±0.9 | 18.4±2.0 | 7.3±1.4 | 6.7±3.6 | 186.7±1.5 | 186.9±1.5 |
| Mistral-7B-V0 | 68.8±2.0 | 17.1±4.3 | 6.5±2.8 | 16.8±2.3 | 179.1±7.7 | 179.4±7.7 |
| Mistral-7B-V1 | 95.9±1.6 | 24.9±0.9 | 10.0±3.8 | 14.3±2.8 | 175.4±6.4 | 175.9±6.2 |
| Mistral-7B-V2 (Ours) | **99.5±0.9** | **31.4±0.5** | **16.0±0.5** | **17.0±0.9** | **169.0±4.3** | **169.4±4.3** |

Appendix A on a concrete example, where UAutomizer itself, the base model, and the V0/V1 stages produce verbose or incorrect invariants, while V2 discovers a compact, correct, and sufficient invariant yielding a $39.75\times$ end-to-end speedup.

*Table 5.* Ablation results on the Easy Split ($n = 239$). **Bold**: best per model family.

| Model | $R_{valid}$ (%) | $R_{correct}$ (%) |
|---|---|---|
| Qwen3-4B (Base) | 100.0±0.0 | 42.3±1.1 |
| Qwen3-4B-V0 | 85.1±2.5 | 26.9±3.8 |
| Qwen3-4B-V1 | 99.0±0.6 | 34.5±1.5 |
| Qwen3-4B-V2 (Ours) | **100.0±0.0** | **50.8±0.9** |
| Mistral-7B (Base) | 96.5±0.6 | 25.8±3.1 |
| Mistral-7B-V0 | 78.0±1.7 | 19.5±0.4 |
| Mistral-7B-V1 | 98.2±0.2 | 34.6±1.3 |
| Mistral-7B-V2 (Ours) | **99.7±0.2** | **41.7±1.7** |
| Llama3.1-8B (Base) | 99.2±0.4 | 13.4±0.4 |
| Llama3.1-8B-V0 | 91.1±1.5 | 31.2±2.7 |
| Llama3.1-8B-V1 | 99.7±0.2 | 41.3±1.7 |
| Llama3.1-8B-V2 (Ours) | **99.7±0.2** | **57.6±2.1** |

**Easy split.** Table 5 shows easy-split results. V2 lifts correctness across all families, with *Llama-3.1-8B-V2* reaching $57.6\%$ and *Qwen3-4B-V2* reaching $50.8\%$. The V0 drop and V1 partial recovery pattern is consistent across all three families, matching the hard split. We report only validation and correctness here as the average direct verification time is already very short ($\sim 6.15\,s$) on this split.

## 7. Limitations & Future Work

WONDA currently relies on UAutomizer as its sole source of training invariants, limiting coverage to solver-verifiable programs and risking inheriting the solver's biases. Extending WONDA to additional backend verifiers is a natural next step. Beyond the training data itself, our current setup neither exposes intermediate reasoning steps nor leverages verifier feedback. Promising directions include chain-of-thought supervision, reinforcement learning driven by verification outcomes, and integrating WONDA-trained models into iterative, counterexample-guided refinement loops.

## 8. Conclusion

We introduced WONDA, a novel data curation pipeline that transforms raw verifier-generated invariants into compact, high-quality training signals with provable quality guarantees. Our results show that data quality, not model scale alone, is a key bottleneck for neural invariant generation: fine-tuning small models on WONDA-curated data substantially improves both invariant correctness and verification speedup across the Qwen3, Llama-3.1, and Mistral families. Most notably, a 4B model surpasses a $20\times$ larger model, and our best 14B model matches frontier models such as GPT-5.2 on end-to-end verification time, without test-time compute overhead. These findings suggest that careful data curation is a practical path to integrating small language models into traditional verifiers, making program verification faster and more accessible.

## Impact Statement

This work contributes to our understanding of how to curate training data for improving model performance on logical reasoning tasks. In addition, this work helps make formal verification more practical in high-stakes domains where the reliability of the software system is important. We do not anticipate significant negative societal impacts arising from this work.

## Acknowledgments

The work of Pinto, Elboher and Katz was partially funded by the European Union (RobustifAI project, ID 101212818). Views and opinions expressed are however those of the author(s) only and do not necessarily reflect those of the European Union or the European Health and Digital Executive Agency (HADEA). Neither the European Union nor the granting authority can be held responsible for them. The work of Wu is partially supported by a gift from the VMware University Research Fund.

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

# Appendix

# A. Additional Examples

```
cohendiv_ll_valuebound50_6.c          Baseline: 214.28 s
void assert(int cond) {
  if (!(cond)) { ERROR: reach_error(); }
}
void assume(int cond) {
  if (!cond) { abort(); }
}
int main() {
  int x, y;
  long long q, r, a, b;
  x = __VERIFIER_nondet_int();
  assume((x >= 0) && (x <= 50));
  y = __VERIFIER_nondet_int();
  assume((y >= 0) && (y <= 50));
  assume(y >= 1);
  q = 0; r = x;
  a = 0; b = 0;
  while (1) {
    if (!(r >= y)) break;
    a = 1; b = y;
    while (1) {
      // Location l ← target
      if (!(r >= (2 * b))) break;
      assert(r >= ((2 * y) * a));
      a = 2 * a;
      b = 2 * b;
    }
    r = r - b;
    q = q + a;
  }
  return 0;
}
```

```
UAutomizer Generated Invariant     18 disjuncts
(((long long) y * 4) == b) &&
((__int128) r + 293) <=
 ((__int128) b * 1024))
&& ((r < (((long long) y * 8) + 1))
 || (r < (((long long) 2 * y) + 1))))
&& (y <= b)) && (a == 4))
|| (((r <= 50) && (1 <= y))
 && (b == ((long long) 16 * y)))
 && (16 == a))
|| ((((b == ((long long) 16 * y))
 && ((16 + (16 * ((r >= 0)
  ? (r / 32) : ((r / 32) - 1))))
 <= b)) && (16 == a))
&& ((__int128) 1998 + r)
 <= ((__int128) b * 1024)))
:
:                        (15 more disjuncts)
```

```
(((((__int128) (((long long) a * 2)
 * y) + (__int128) 2 * (((__int128)
((long long) a * y * 2) * a))) == (b
 + (__int128) ((long long) b * 2))))
 || (((((__int128) ((long long) a *
y * 2) + (long long) a * y) == (b +
(__int128) ((long long) b * 3))) &&
(((__int128) ((long long) b * 4) +
b) + (((__int128) ((long long) a *
y * 2) * a)))))) || (((((a >= 0) ?
(a % 2) :  ((a % 2) + 2)) != 0) &&
(((__int128) ((long long) a * y * 2)
< (b + (((__int128) ((long long) a *
y * 2) * a)))))))
```
**Qwen3-8B-V0**

**Incorrect**

```
(r >= (2 * y) * a) &&
(b == (2^(a / 2)))
```
**Qwen3-8B Base**

**Incorrect**

```
a == 0 || b < y + 1
```
**Qwen3-8B-V1**

**Incorrect**

```
a * y == b
```
**Qwen3-8B-V2**

**Correct, Sufficient (39.75×)**

*Figure 5.* Inference-time invariant predictions on `cohendiv_ll_valuebound50_6.c`. The task is to synthesize an inductive invariant at the inner loop entry point (`// Location l`). The UAutomizer-generated invariant (top right) enumerates 18 disjuncts, each encoding `b == k*y` and `a == k` for specific powers of two; the compact closed-form `a*y == b` captures all of them. V2 discovers this generalization, reducing verification from 214.28 s to 5.39 s (39.75× speedup including LLM inference), while earlier stages produce incorrect or obfuscated invariants.

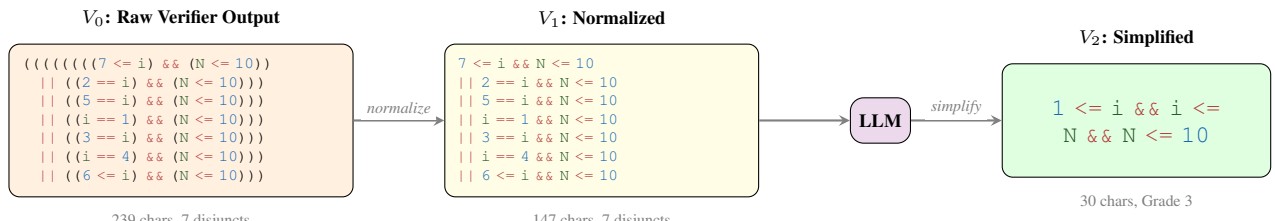

*Figure 6.* WONDA pipeline. The LLM factors out the common constraint `N <= 10` and reduces case enumeration into a compact range, achieving a **3.47x** verification speedup.

# B. Prompts

## B.1. Prompt for training and evaluation

We provide here the system and user prompts for loop invariant generation task, used for both training and evaluation.

```
System:
    You are an expert C programmer and highly proficient
    in generating strong loop invariants
    for C programs that accelerate traditional verifiers' verification process.

    ## Input format
    - A C program instrumented with loop markers of the form:
      '''c
      INVARIANT_MARKER_k(); // appears at the *start of each loop body*
      '''
      - The program contains a single target property as an assertion of the form:
      '''c
      assert(<target_property>);
      '''
    - A target loop marker (e.g., "INVARIANT_MARKER_1")

    ## Task
    - Propose ONE loop invariant that is intended to hold specifically at the target loop
        marker.
    - The invariant should help prove the target property and be inductive if possible.

    ## Output format
    - Output MUST be a single JSON object on one line wrapped in '''json''' tags and
        nothing else.
    - The JSON MUST have exactly these keys:
      - "marker": MUST be exactly the target loop marker (e.g., "INVARIANT_MARKER_1")
      - "content": ONLY a valid C boolean expression for the invariant.

    ## Output format example
    '''json
    {"marker":"<target_marker>","content":"<content>"}
    '''

User:
  ## User Input
  ### C Program
  '''c
  {program}
  '''
  ### Target Loop Marker
  {target_marker}
```

## B.2. Prompt for Invariant Simplification

We provide here the full system and user prompts used for the simplification step in the WONDA pipeline.

```
System:
   ## Task
   Given the C program and the invariant, your task is to simplify the
   invariant to a more compact and general form.

   ## Output format
   - Output MUST be a single JSON object.
   - The JSON MUST have exactly these keys:
     - "simplified_invariant": A single compact, inductive, C boolean
       expression, nothing else.
     - "rationale": A short explanation of why you simplified the
       invariant to the given form.
   ## Output format example
   {"simplified_invariant":"<simplified_invariant>",
       "rationale":"<rationale>"}

   ## Guidelines
   - The simplified invariant should be logically weaker than (or
     equivalent to) the original, but still inductive and strong enough
     to prove the target property.
   - Prefer LINEAR arithmetic expressions (the verifier struggles with
     non-linear math like x*y)
   - Prefer mathematical relationships over case enumeration
   - Look for patterns across disjuncts (e.g., repeated structure with
     varying constants)
   - Generalize enumerated values to ranges (e.g., "i == 1 || i == 2
     || i == 3" -> "1 <= i && i <= 3")
   - Remove tautological constraints (e.g., "a == a", "n <= n",
     "0 <= 0", "a + 0 == a", "true", "1")
   - Remove constraints on constant variables (variables initialized
     but never modified in loops)
   - Replace redundant constraints with simpler equivalents (e.g.,
     "a <= b && b <= a" -> "a == b")
   - Ensure the simplified invariant is still inductive (holds before
     loop and preserved by each iteration)
   - Use the program context to understand variable semantics and
     loop structure
   - Use ONLY plain ASCII characters in your output (no Unicode symbols)

User:
   Simplify the following invariant for the given C program and marker.
   c_program:
   ```c
   {program}
   ```
   invariant:
   ```c
   {invariant}
   ```

   marker:
   ```c
   {marker}
```
```

# C. Algorithms

---

**Algorithm 1** Candidate Invariant Grading

---

1: **Input:** Verification query $\langle A, P, q \rangle$, location $l$, candidate predicate $\varphi$, baseline time $t_b$
2: **Output:** Quality grade $g \in \{0, 1, 2, 3\}$
3:
4: **if not** SYNTAXVALID($\varphi$) **then**
5:      **return** 0 {Invalid syntax}
6: **end if**
7:
8: $I \leftarrow (l, \varphi)$
9: {Parallel execution (cf. Figure 2)}
10: $(\mathcal{V}_1, t_1) \leftarrow \mathcal{V}(A, P, I)$ {Correctness Check}
11: $(\mathcal{V}_2, t_2) \leftarrow \mathcal{V}(A \cup \{I\}, P, q)$ {Sufficiency Check}
12: $t_v \leftarrow \max(t_1, t_2)$ {Total wall-clock time}
13:
14: **if** $\mathcal{V}_1 \neq$ TRUE **then**
15:      **return** 0 {Incorrect (not inductive)}
16: **else if** $\mathcal{V}_2 \neq$ TRUE **then**
17:      **return** 1 {Correct but not sufficient}
18: **else if** $t_v \geq t_b$ **then**
19:      **return** 2 {Correct and sufficient, but no speedup}
20: **else**
21:      **return** 3 {Correct, sufficient, and provides speedup}
22: **end if**

---

**Algorithm 2** Invariant Simplification

---

1: **Input:** Verification query $\langle A, P, q \rangle$, location $l$, normalized invariant $\varphi_{\text{norm}}$, baseline time $t_b$, $N$ candidates, minimum character length $\eta$
2: **Output:** Set $R$ of qualifying simplified invariants with their corresponding grades.
3: $R \leftarrow \emptyset$
4: **if** $\varphi_{\text{norm}} \in \{0, 1\}$ **then**
5:      **return** $R$
6: **end if**
7: **if** $|\varphi_{\text{norm}}| > \eta$ **then**
8:      $\mathbf{C} \leftarrow$ LLM($P, \varphi_{\text{norm}}, l, N$)
9:      $\mathbf{C} \leftarrow$ DEDUPLICATE($\mathbf{C}$)
10:      **for each** $\varphi \in \mathbf{C}$ **do**
11:          **if** $\varphi \in \{0, 1\}$ **then**
12:              **continue**
13:          **end if**
14:          $g \leftarrow$ GRADECANDIDATE($\langle A, P, q \rangle, l, \varphi, t_b$) {via Alg. 1}
15:          **if** $g \geq 2$ **then**
16:              $R \leftarrow R \cup \{(\varphi, g)\}$
17:          **end if**
18:      **end for**
19: **end if**
20: **if** $R = \emptyset$ **then**
21:      $g \leftarrow$ GRADECANDIDATE($\langle A, P, q \rangle, l, \varphi_{\text{norm}}, t_b$)
22:      **if** $g \geq 2$ **then**
23:          $R \leftarrow \{(\varphi_{\text{norm}}, g)\}$
24:      **end if**
25: **end if**
26: **return** $R$

---

## D. Training Data Statistics

**Pipeline Yield.** We applied WONDA to 4,000 raw verifier-generated invariants. Of these, 3,932 (98.3%) produced at least one accepted candidate ($G(\varphi) > 0$); only 68 (1.7%) yielded an empty set. Of the 2,995 verbose invariants ($|\varphi_{\text{norm}}| \geq 20$), the LLM simplification stage generated 11,980 candidates ($N{=}4$), retaining 6,584 (55.0%) with $G(\varphi) \geq 2$, along with 187 $G(\varphi){=}1$ and 39 normalized fallbacks. The remaining 1,005 compact invariants were verified as-is; 983 (97.8%) yielded a $G(\varphi) \geq 2$ candidate. The pipeline produces 7,763 curated rows in total; restricting to $G(\varphi) \geq 2$ and dropping examples whose full prompt-completion sequence (after applying the chat template) exceeds 1,024 tokens gives the final *V2* set of 7,284 samples.

**Dataset Statistics.** As shown in Figure 7, the *V2* set is partitioned 80/20 into 5,827 training and 1,457 validation samples, comprising two quality tiers: Correct and Sufficient ($G(\varphi) = 2$, 4,516 samples) and Provides Speedup ($G(\varphi) = 3$, 2,768 samples). The mean sequence length is 518 tokens; generated invariants are highly concise at 15.8 tokens on average. The $G(\varphi) = 3$ subset achieves a mean speedup of $2.13\times$ with peaks up to $41.39\times$, demonstrating that WONDA-curated invariants offer significant computational advantages for the formal solver.

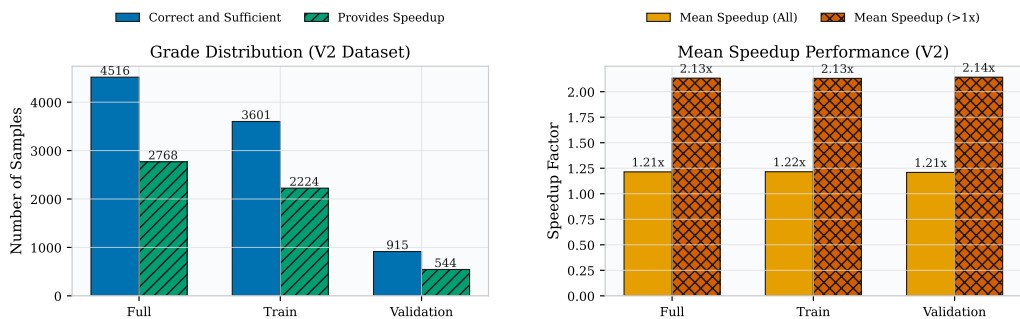

*Figure 7.* Statistical distribution of the *V2* curated dataset across the 80/20 train-validation split.

## E. Hard Split Evaluation

To quantify the difficulty of the hard cases, we evaluate the baseline verifier on this subset 3 times and use the median timing among them. Figure 8 (Left) reports the verifier's performance on the 123 selected hard instances. The verifier fails to solve 16.3% of these cases (compared to 5.5% across all instances), confirming that these represent substantially more challenging verification problems. Figure 8 (Right) shows the runtime distribution of the baseline verifier on the 123 hard cases. Execution times range from 15.5 s to the 600 s timeout, with a median of 112.2 s and a mean of 193.0 s.

Hard Split

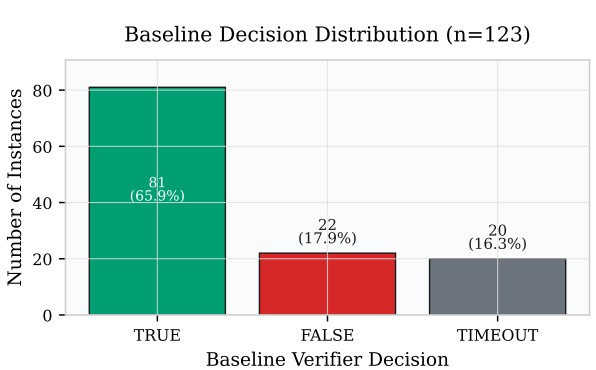
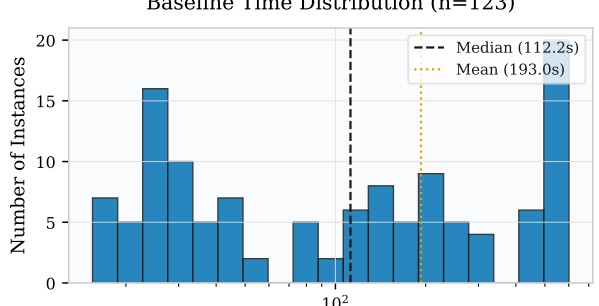

*Figure 8.* **Hard evaluation split baseline characterization.** 72 programs expanded to 123 per-loop instances (`median_timing>15 s`). **(a)** Baseline verifier decisions ($n{=}123$ instances). **(b)** Baseline time distribution ($n{=}123$; instance-level baseline VBP).

## E.1. Benchmark Characterization

The hard evaluation split comprises 72 SV-COMP programs (which expands to 123 per-loop instances). Figure 9 summarizes the code structure:

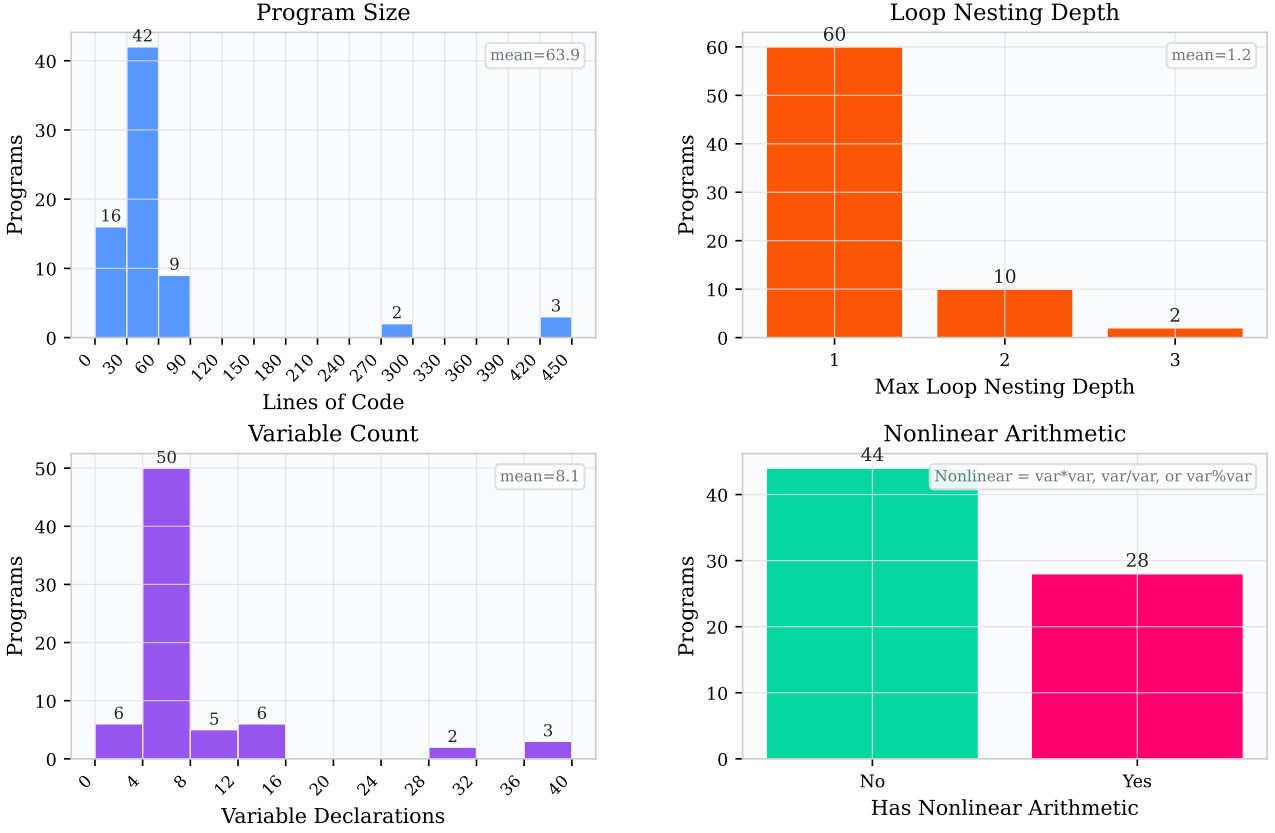

*Figure 9.* **Benchmark characterization on hard programs** ($n$=72 unique C programs).

## E.2. Post-hoc Timeout Sweep

We performed a post-hoc timeout sweep ($T \in \{15, 30, \ldots, 600\}$ s), shown in Figure 10. At each $T$, rates count only instances that finish within $T$.

WONDA-V2 dominates its base counterpart at every timeout on all three panels. $R_{\text{correct}}$ and $R_{\text{speedup}}$ rise with $T$ for both, but V2 climbs faster and plateaus substantially higher; gains are already visible at short timeouts. $\text{VBP}_{\text{E2E}}$ stays lower for V2 throughout, while base models remain near the $\sim$193 s verifier baseline.

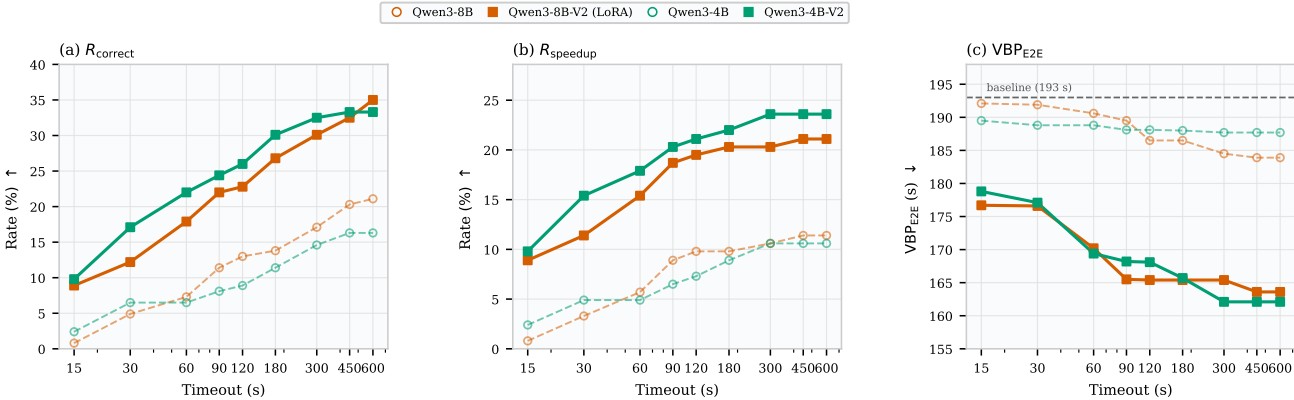

*Figure 10.* **Post-hoc timeout sweep on hard instances** ($n{=}123$; $T \in \{15, 30, \ldots, 600\}$ s). **(a)** $R_{\text{correct}}$, **(b)** $R_{\text{speedup}}$, and **(c)** $\text{VBP}_{\text{E2E}}$ vs. $T$ for Qwen3-4B/8B base (circles, dashed) and WONDA-V2 (squares, solid; 8B-V2 uses LoRA). Dashed line in (c): solver baseline ($\approx$193 s).

## F. Model training details

*Table 6.* Full SFT hyperparameters for all other fine-tuned models on invariant generation (no-think mode).

| Hyperparameter | Qwen3-0.6B / 4B | Mistral-7B / Llama3.1-8B | Qwen3-14B |
|---|---|---|---|
| Learning Rate | $1 \times 10^{-4}$ | $2 \times 10^{-5}$ | $5 \times 10^{-5}$ |
| LR Scheduler | cosine_with_min_lr (min ratio: 0.1) | cosine | cosine_with_min_lr (min ratio: 0.1) |
| Warmup Ratio | 0.03 | 0.05 | 0.03 |
| Epochs | 2 | 3 | 3 |
| Effective Batch Size | 32 | 32 | 32 |
| Max Seq. Length | 1024 | 1024 | 1024 |

*Table 7.* Supervised Fine-Tuning (SFT) Hyperparameters for Qwen3-8B-V2 on invariant generation with LoRA (Hu et al., 2022).

| Category | Hyperparameter | Value / Setting |
|---|---|---|
| *Optimizer & LR* | Learning Rate | $5 \times 10^{-4}$ |
| | LR Scheduler | cosine_with_min_lr (min ratio: 0.1) |
| | Warmup Ratio | 0.03 |
| *Batch Config* | Epochs | 2 |
| | Effective Batch Size | 32 |
| | Max Seq. Length | 1024 tokens |
| *LoRA (PEFT)* | Rank ($r$) | 128 |
| | Alpha ($\alpha$) | 64 |
| | Dropout | 0.05 |
| | Target Modules | All linear layers + embed_tokens |

*(a)* Qwen3 non-think models (0.6B–14B).

| Hyperparameter | Value |
|---|---|
| Max New Tokens | 1024 |
| Temperature | 0.7 |
| Top-$p$ | 0.8 |
| Top-$k$ | 20 |
| Min-$p$ | 0 |
| Repetition Penalty | 1.1 |

*(b)* Mistral-7B-Instruct-v0.3 and Llama-3.1-8B-Instruct.

| Hyperparameter | Mistral-7B Instruct | Llama-3.1-8B Instruct |
|---|---|---|
| Max New Tokens | 1024 | 1024 |
| Temperature | 0.6 | 0.6 |
| Top-$p$ | 0.9 | 0.9 |

*Table 8.* Sampling hyperparameters used for each model.

## G. Hardware, UAutomizer Release & Configuration

We used UAutomizer release for SVCOMP-2025 (Heizmann et al., 2013) with the configuration shown in Table 9.

*Table 9.* UAutomizer Configuration

| Parameter | Value |
|---|---|
| Version | 0.3.0-dev-d790fec[1] (Java 21, jdk-21.0.1) |
| Property | `unreach-call.prp` |
| Property specification | `CHECK( init(main()), LTL(G ! call(reach_error())) )` |
| Architecture | 32-bit |
| Memory limit | 16 GB (enforced via `runlim`[2]) |
| Timeout | 600 s per verification task |

**Hardware.** Experiments ran on a Linux SLURM cluster node with 8 AMD EPYC 9354 cores, 256 GB RAM, and one NVIDIA L40S GPU. The verifier was memory-limited to 16 GB using `runlim`.

---

[1]`https://zenodo.org/records/14209043`
[2]`https://github.com/arminbiere/runlim`

