# OpenReview forum: "Not All Invariants Are Equal: Curating Training Data to Accelerate Program Verification with SLMs"
_ICML.cc/2026/Conference — ICML 2026 regular_

### Official Review · Reviewer_Ga8f · 2026-03-10

**Soundness:** 2
**Presentation:** 3
**Significance:** 2
**Originality:** 3
**Overall Recommendation:** 4
**Confidence:** 3

**Summary:**

This paper tackles loop invariant generation. Typically, training data is generated by directly using the output from loop invariant generation tools. However, the output of these tools is often extremely un-idiomatic and difficult to understand - a common problem with purely symbolic tools. The authors use a two step approach to simplify the output - an AST based normalization and then an LLM performing a set of rewrites that preserve the semantics of the invariant. These simplified examples are then used to generate a dataset which is used to train small language models.  Their results show an improvement in several of the Qwen models.

**Compliance With Llm Reviewing Policy:**

Affirmed.

**Final Justification:**

This paper is VERY borderline for me.

A big reason it's borderline as-is is because I think there's some overpromising in the introduction, which I find a little frustrating. A big issue in using symbolic tools to generate training data is that anything that the symbolic tool cannot solve isn't included in the training data, and so the model has trouble outperforming the symbolic "teacher". The authors talk about this issue in the introduction, but it's not really addressed/fixed through the paper IMO because they are just modifying the outputs of the symbolic tools to increase the pedagogical value. Invariants generated from the LLM's might be more idiomatic, but they don't meaningfully surpass the "teacher".

However, I think this paper does have some notable merits. I think their method of making these invariants more pedagogically friendly is clever, and their approach clearly helps performance of LLM's trained on invariants. Even though I'm not convinced that LLM's trained on invariants are better than symbolic tools yet, I do think WONDA makes a meaningful step in that direction.

I think the actual technical contribution of the paper just barely passes past the acceptance bar for me. And I think if the authors were able to submit new drafts, they could fix the over-promising to the point where the writing passes the acceptance bar for me as well. They acknowledged in their rebuttals that it does not completely escape the teacher-ceiling trap. So I'll keep my score for this paper at a weak accept.

**Key Questions For Authors:**

The main questions I have are the three weaknesses:

1. It's not clear to me how the "overfitting trap" mentioned in the introduction is addressed by this approach. How is this addressed?
2. I think an ablation with training directly on the un-normalized invariants is important - that shows that it's WONDA that's making the improvements and not just the training itself. Have the authors done this?
3. Would this approach generalize to other models besides Qwen?

I think if 1 and 2 are addressed appropriately I can raise my score to an accept.

**Limitations:**

No, there is no limitations section. Discussing the limitations of this approach in general would be good.

**Strengths And Weaknesses:**

#### Strengths
I think this is a good approach. I think using an LLM to perform a set of deterministic rewrites to make things idiomatic without losing the semantics is an intuitive and clever approach. The results show an improvement.

#### Weaknesses
I think the evaluation is a little weak. I think it's missing 3 things (in order of importance):
1. It's not clear to me how the "overfitting trap" mentioned in the introduction is addressed by this approach.
2. I think an ablation with training directly on the un-normalized invariants is important - that shows that it's WONDA that's making the improvements and not just the training itself.
3. Would this approach generalize to other models besides Qwen?
I also think a limitations section would be good.


#### Overall
I think this paper has some good ideas and a decent evaluation. It's just missing an ablation and more discussion of the overfitting trap mentioned in the introduction.

---

> ### Author Rebuttal · Authors · 2026-03-31
>
> We thank the reviewer for the constructive evaluation.
>
> **Q1.** Thank you for this question. The "overfitting trap" occurs if a model is trained exclusively on the outputs of the symbolic solver, which we seek to improve. To address this issue, *our key insight is to create training data that neither the symbolic solver nor the language model could produce on its own*. To do so, we begin with raw, verbose, and often ungeneralized invariants produced by the symbolic solver. We then utilize a combination of analytic rules and LLM-based rewriting to transform these raw invariants into elegant, more insightful invariants. These rewritten versions are verified by the symbolic solvers. This process results in novel and mathematically sound invariants used to fine-tune the model, which empirically resulted in models that can propose insightful invariants for complex programs that would have remained out of reach for the symbolic solver on its own.
>
> **Q2.** We agree. And the V0 configuration is such ablation (See Table 4 in the paper): it trains on raw, un-normalized UAutomizer invariants, isolating the effect of naive SFT without WONDA's curation. To further strength our claim, we add a new ablation study on Qwen3-4B model (Full Fine-tuning) in [Table T3](https://tinyurl.com/47uhmfv9) which shows a similar trend as other scales and will be added to Table 4 in the final version.
>
> **Q3.** To address this, we have fully finetuned LLaMA 3.1-8B-Instruct on the WONDA V2 dataset:
>
> | Model | Valid. | Correct. | Speedup | VBP (s) |
> |---|---|---|---|---|
> | LLaMA 3.1-8B (Base) | 96.7% | 19.5% | 8.9% | 182.0 |
> | **LLaMA 3.1-8B-V2 (ours)** | **100%** | **44.7%** | **17.9%** | **178.2** |
>
> The +25.2pp gain in R_correct and the +9pp in speedup matches the magnitude observed on Qwen3-8B at the same scale.
>
> **Limitations.** In response to the reviewer's request, we will add a detailed limitations section in the revision.

---

> > ### Author Rebuttal · Reviewer_Ga8f · 2026-03-31
> >
> > I see. I'm still a bit confused on the overfitting trap. If I understand, the pipeline in WONDA makes the invariants more *idiomatic*, but they're still modifications of the original invariant generated by UAtomizer right? If there is a loop where UAtomizer is *incapable* of generating an invariant for it, will that still be handled by WONDA? Can a model trained with WONDA *surpass* the original verifier used (in this case, UAutomizer)? Importantly, do the models trained with WONDA outperform UAutomizer on the final evaluation dataset?
> >
> > > empirically resulted in models that can propose insightful invariants for complex programs that would have remained out of reach for the symbolic solver on its own.
> >
> > Is this in the paper? Did the models generate invariants that were unable to be generated by UAutomizer/did the models outperform UAutomizer?

---

> > > ### Author Response · Authors · 2026-04-02
> > >
> > > > If I understand, the pipeline in WONDA makes the invariants more idiomatic, but they're still modifications of the original invariant generated by UAutomizer right?
> > >
> > > Yes. We start from invariants generated by UAutomizer, which are analytically normalized and then provided as hints for the LLM. The model uses these as a starting point to reason about and improve upon. Importantly, the resulting modifications can be quite substantial (see Fig. 2 in the paper), going well beyond minor syntactic changes.
> > >
> > > > If there is a loop where UAutomizer is incapable of generating an invariant for it, will that still be handled by WONDA?
> > > WONDA assumes access to an initial invariant from UAutomizer. In practice, all programs in the InvBench training dataset can be solved by UAutomizer.
> > >
> > > The effect of WONDA is that it improves the pedagogical value of the invariants. We acknowledge that this setup does not fully escape the dependency on an analytic invariant generator. To be truly independent of the invariant generator, we believe RL is necessary, as it removes the reliance on such generators. We view WONDA as a necessary foundation for RL-based approaches (as discussed in our response to reviewer aprm). We will expand this discussion in the revised version.
> > >
> > > > Can a model trained with WONDA surpass the original verifier used (in this case, UAutomizer)? Importantly, do the models trained with WONDA outperform UAutomizer on the final evaluation dataset? "empirically resulted in models that can propose insightful invariants for complex programs that would have remained out of reach for the symbolic solver on its own." Is this in the paper? Did the models generate invariants that were unable to be generated by UAutomizer/did the models outperform UAutomizer?
> > >
> > > Yes. Our experiments show that WONDA-trained models generate invariants that improve upon UAutomizer:
> > > - On average, model-generated invariants reduce solving time on the hard evaluation set (193s → 165.5s VBP).
> > > - More importantly, on the evaluation set (where UAutomizer does not solve all instances), the model generates invariants
> > > that enable solving previously unsolved programs. See the Solved column in Table 3: Qwen3-4B-V2 resolves 2-3 of the 20 UAutomizer timeouts per run. Notably, Qwen3-8B-V2 (LoRA) solved 4 timeout instances in a single run, which exceeds the best single-run result of both GPT-5.2 and GPT-OSS-120B (3 each).
> > >
> > >
> > > Since InvBench is constructed such that UAutomizer solves all training instances but not all testing instances, this result suggests that WONDA-trained models exhibit meaningful generalization beyond the original solver's capabilities.
> > >
> > >
> > > We have added to our paper [a concrete example](https://imgland.net/i/tshbxsAk/example_of_wanda_invariants.png) illustrating this. On the benchmark ```cohendiv_ll_valuebound50_6.c```, Qwen3-8B-V0 (trained on raw data) generated an incorrect, obfuscated invariant, while Qwen3-8B-V2 generated the insightful constraint ```a*y == b```, reducing total verification runtime from 214s to 5.39s (a 39.75× end-to-end speedup). For comparison, a plain UAutomizer proof yielded [these loop invariants](https://imgland.net/i/BDWegrgD/carbon-20.png), which are vastly more complex and uninterpretable.
> > >
> > > We will add additional examples in the revised version and hope that the above clarifications address the reviewer's concerns. We are happy to discuss further.

---

### Official Review · Reviewer_4GrS · 2026-03-10

**Soundness:** 3
**Presentation:** 3
**Significance:** 3
**Originality:** 2
**Overall Recommendation:** 4
**Confidence:** 4

**Summary:**

The scope of this paper is the application of neural language models to loop invariant generation, an aspect of program verification. The authors note that traditional (symbolic) and modern (neural) invariant generation tools produce non-minimal invariants that are inefficient from a runtime standpoint and hard to leard. To fill this gap, the paper demonstrates a two-stage data pipeline ("Wonda") that produces high-quality invariants which are then used as training data for small language models, which in turn demonstrate comparable performance to much larger general-purpose LLMs.

**Compliance With Llm Reviewing Policy:**

Affirmed.

**Final Justification:**

In light of the new evidence provided by the authors in their rebuttals, I have increased the Soundness score.

**Key Questions For Authors:**

1. The first part of the Wonda pipeline seems to be designed to address shortcomings of (symbolic) invariant generators and has to normalize the AST with a series of syntax transformation passes. Why did the authors choose UAtomizer as a starting point even though it seemingly does not produce minimal invariants?  Did the authors experiment with various proportions of syntactic vs neural rewriting, and what are their observations ?

2. It's not clear (to me at least) how this work positions itself with respect to the most related literature cited therein (namely Pei 2023 and Wei 2025) (Section 1, 3rd paragraph). Some discussion on the relationship with those papers would be helpful.

3. Did the authors experiment with alternative prompts e..g with a varying amount of examples in each ?

**Limitations:**

yes

**Strengths And Weaknesses:**

Soundness: The claims in the paper are mostly well supported, but in my opinion Claim 2 (line 99) needs a bit more work to be convincing: it would be good to know what is the rejection rate of Algorithm 1 (since it can produce an empty set in some cases). The Wonda pipeline also hinges on a LLM for rewriting candidates, therefore the quality of the V2 data is sensitive to its prompt as well; even though Kimi K2 is generally smart, some prompt ablations (and in particular, moving some simpler LLM rewriting tasks like range generalization or constraint factoring to the syntactic rewriting stage) could clarify the relative strength of the Wonda approach.

Presentation: The paper is well written and takes the time to introduce all relevant definitions (perhaps to the detriment of the results section, but this is justifiable as ICML is not a program verification conference). The algorithmic pipeline is spelled out in sufficient detail.

Significance: The topic is somewhat niche, but the end result is encouraging enough: moderately-sized (600M) language models can be fine tuned to produce efficient loop invariants with this method, which could have interesting downstream applications in IDEs.

Originality: The paper offers a novel combination of existing techniques in a well-explored area. The originality aspect would also be improved by some targeted ablations (as described above).

---

> ### Author Rebuttal · Authors · 2026-03-31
>
> We thank the reviewer for the thoughtful evaluation.
>
> **W1.** Of 4,000 input invariants, **3,932** (98.3%) produced at least one qualifying output. Only **68** (1.7%) yielded an empty set, broken down as follows:
> **Verbose invariants** (normalized length >= 20, sent to LLM with N=4): 2,995 total. 11,980 LLM candidates were generated; **6,650** (55.5%) passed verification. The 44.5% per-candidate failure rate is expected (the verifier rejects incorrect simplifications), but because N=4, most invariants still get at least one passing candidate. **55** verbose invariants yielded an empty set (all 4 LLM candidates and the normalized fallback failed verification within the 600s timeout).
> **Compact invariants** (normalized length < 20, kept as-is after normalization): 1,005 total. 992 (98.7%) passed verification; 13 failed.
> The 68 empty-set cases (55 verbose + 13 compact) are programs where verification exceeds the 600s timeout regardless of invariant form. The pipeline outputs 7,763 entries from accepted invariants; filtering to grade >= 2 retains **7,284 SFT training samples**. We will add these statistics to the revision.
>
> **W2 / KQ3.** The pipeline uses a zero-shot prompt while the verifier serves as a filter: a weaker prompt increases the rejection rate but cannot degrade output quality, since no unverified invariant enters the training set. The current prompt achieves a 55.5% per-candidate pass rate, and the V1-to-V2 ablation confirms the LLM simplification step contributes meaningful gains at every model scale (Table 4). Prompt optimization could further improve the per-candidate pass rate and shift the grade distribution toward more speedup-providing samples; we plan to explore this in future work.
>
> **KQ1.** UAutomizer is the reigning SV-COMP champion (1st place, ReachSafety 2024-2025) with the broadest coverage of any public verifier. Non-minimality is universal to CEGAR-based verifiers; WONDA addresses this limitation regardless of the upstream source. We did not vary the boundary between syntactic and neural stages; the split is fixed by design (Stage 1 handles equivalence-preserving rewrites, Stage 2 handles semantic transformations requiring verification). The V0/V1/V2 ablation study in Table 4 and the newly added Qwen3-4B ablation in [Table T3](https://tinyurl.com/47uhmfv9) isolate each stage's marginal contribution under the current design.
>
> **KQ2.** The three works are complementary. Pei et al. fine-tune on invariants from **Daikon** (dynamic analysis), i.e. template-based expressions that are already clean but, importantly, lack formal correctness guarantees. Wei et al. (InvBench) propose a verifier-based evaluation framework with formal soundness and fine-tune on raw solver outputs, equivalent to WONDA's V0. WONDA addresses the **data quality** gap: we adapt Wei et al.'s evaluation framework and show that static verifiers produce correct but obfuscated invariants, which WONDA's pipeline transforms into idiomatic training targets. The V0-to-V2 improvement quantifies the value of WONDA over the raw-data approach of both prior works. We will add this positioning to Related Work.

---

> > ### Author Rebuttal · Reviewer_4GrS · 2026-04-02
> >
> > Thank you to the authors for the thoughtful discussion. Provided the above remarks will make it to the final version, my concerns with this work will have been put to rest.

---

> > > ### Author Response · Authors · 2026-04-02
> > >
> > > Thank you for the feedback. As ICML does not allow us to upload revisions, we will incorporate all clarifications and additional results discussed here in the final version. We would appreciate it if you would consider adjusting your scores. Please do let us know if you have any additional questions or concerns!

---

### Official Review · Reviewer_76Vk · 2026-03-13

**Soundness:** 2
**Presentation:** 3
**Significance:** 3
**Originality:** 3
**Overall Recommendation:** 3
**Confidence:** 4

**Summary:**

This paper addresses the problem of loop invariant synthesis for automated program verification. The authors propose WONDA, a data curation pipeline that combines AST-based normalization with LLM-driven semantic rewriting to transform raw, solver-generated invariants into high-quality training data. Fine-tuning SLMs on this curated data yields significant performance improvements.

**Compliance With Llm Reviewing Policy:**

Affirmed.

**Final Justification:**

Invariant inference is an important problem and the curation pipeline is a solid contribution. However, I am not fully convinced by the arguments around refinement, some simple experiments with iterative feedback would have gone a long way in demonstrating whether the models are genuinely learning to reason about invariants or pattern-matching to the training distribution. The ~9% correctness drop for larger models also needs to be studied in more detail to understand what WONDA is truly unlocking versus potentially overwriting. Maintaining my current score.

**Key Questions For Authors:**

1. How does the performance of the tool vary with timeout?  A plot of verification performance vs. timeout would help clarify whether the gains are robust or concentrated in a narrow operating regime.

**Limitations:**

Yes

**Strengths And Weaknesses:**

Strengths:

1. Loop invariant synthesis is a well-known problem in program verification, and the question of how to train smaller, more efficient models for this task is timely and relevant.

2. Isolating the contribution of each pipeline stage (V0 → V1 → V2), demonstrating that LLM-based simplification with verification filtering is informative and well studied

Weakness:
1. The primary focus of this work is on generating invariants faster using a smaller model, and the WONDA curation pipeline is explicitly optimized for compactness and solver speedup. However, in established invariant generation tracks such as SyGuS-INV, correctness is weighted more heavily than speed. A dataset curation paper that claims to improve the quality of training data should demonstrate improvements on both axes, correctness and speed,  rather than treating speedup as the dominant signal. The correctness rates of the fine-tuned SLMs (Qwen3-4B-V2) remain substantially below GPT-5.2 , even as their VBP figures are comparable. This gap deserves more attention. It's important discuss the trade-off between invariant correctness and verification speedup more explicitly
2. The paper evaluates on 123 "hard" instances from SV-COMP but provides limited characterization of the benchmark suite. What is the distribution of program sizes (lines of code), loop nesting depths, and variable counts? What is the distribution of invariant complexity,  are the ground-truth invariants predominantly linear arithmetic, or are non-linear?
3. There is no systematic analysis of why the models fail on hard instances. Understanding why certain instances remain hard would be valuable and help scope the claims of the paper more precisely.
4. Most competitive invariant generation tools operate in a generate → verify → refine loop, using counterexamples from the SMT solver to repair invalid candidates. The proposed system is purely one-shot. This limits the comparison: the relevant question is not just whether the model generates a correct invariant on the first try, but whether it can reason over counterexamples to iteratively repair incorrect candidates. Evaluating the model's capacity for counterexample-guided refinement would be a much stronger test of whether it has genuinely learned the underlying verification logic, rather than pattern-matching to the training distribution.
5. The training and evaluation data both derive from the InvBench suite (SV-COMP programs verified by UAutomizer). It is unclear whether the performance gains generalize to other benchmark families. It would be helpful to evaluate at least one additional benchmark suite,  for example, SyGuS-INV benchmarks or programs from a different language, to assess out-of-distribution generalization.

6. The evaluation metric section is a bit confusing and hard to follow.

---

> ### Author Rebuttal · Authors · 2026-03-31
>
> We thank the reviewer for the feedback.
>
> **KQ1.** We performed a post-hoc timeout sweep (T∈{15,30,…,600}s), shown in [Fig. R1](https://tinyurl.com/2z5yprt6). Regardless of timeout, fine-tuning yields consistent gains in correctness rate, speedup rate, and VBP.
>
> **W1.** We agree that invariant quality is important, and we define it not just by correctness but also usefulness (whether the invariant accelerates verification). WONDA improves both substantially. As the reviewer points out, there is a trade-off: larger reasoning models achieve higher correctness at the cost of model size and inference overhead. [Fig. R2](https://tinyurl.com/4dbrjbz7) illustrates this. 4B-V2 matches OSS on VBP despite lower correctness rate, while without fine-tuning the base model is nowhere near OSS or GPT-5.2. WONDA boosts VBP without increasing inference cost.
>
> **W2 / W3.**
> **Benchmark Characterization ([Fig. R3](https://tinyurl.com/yc3ww66p)).** The 123 Hard instances derive from 72 SV-COMP programs. We clarify the test benchmarks have no ground-truth labelled invariants, so we cannot characterize invariant complexity directly. We report program-level nonlinear arithmetic as a complexity proxy.
>
> **Failure Analysis.** [Fig. R4a](https://tinyurl.com/mr5u8fjz) and [R4b](https://tinyurl.com/3ksrr2fn) show failure modes stratified by program properties for 4B-V0 and 4B-V2. Correctness failure dominates for both, but improvement is broad: nonlinear instances see the largest reduction (85.7%→63.3%), linear instances also improve (78.4%→68.9%), larger programs benefit substantially (LOC≥200: 80%→30%), and medium variable counts improve (10–20 vars: 85.7%→42.9%). Interestingly, V2 shows slight regression on the shortest programs (0–30 LOC), suggesting that WONDA's simplification signal, optimized for complex invariants, may not generalize well to simpler verification tasks. Complexity-aware curriculum training is a promising direction to address this.
>
> **W4.** Counterexample-guided refinement is indeed central to classical invariant generation, and we agree that evaluating a model's capability to refine its previous prediction using counter-examples is a compelling research direction. However, this is an orthogonal direction to our work, where the goal is to evaluate whether careful curation of the training data can improve a language model’s ability to generate high-quality invariants on its own. We believe this goal is also meaningful when it comes to CEGIS-style reasoning, as a better one-shot invariant generator provides better initialization for the CEGIS loop.
>
> Implementing a full counterexample-guided LLM-based loop remains a substantial research challenge. It is still unclear how to distill the feedback from the underlying SMT solver into a digestible form that can provide meaningful guidance to the LLM. For example, Lemur (Wu et al., ICLR 2024), a state-of-the-art LLM-enabled verifier, employs iterative refinement but does not incorporate explicit verifier counterexamples into its feedback loop. We do believe that designing such a CEGIS mechanism and evaluating its interaction with training data quality are promising directions for future work.
>
>
> **W5.** Thank you for this comment. We conduct two additional generalization experiments.
> **Easy split.** [Table T1](https://tinyurl.com/55yxfmf3) shows results over 3 runs on the easy split (239 instances), held out from training and main evaluation. The results confirm the training signal transfers to other unseen programs.
> **SyGuS & Code2Inv.** The original SyGuS benchmarks use SMT-LIB syntax unsupported by our current framework. To still address this, Table T2 shows results on Code2Inv (133 C programs from SyGuS 2017) over 10 runs.
> | Model | R_correct (%) |
> |---|---|
> | Qwen3-0.6B | 22.8±1.7 |
> | Qwen3-0.6B-V2 | **53.5±2.1** |
> ||
> | Qwen3-4B | 66.8±0.7 |
> | Qwen3-4B-V2 | **64.4±1.9** |
> ||
> | Qwen3-8B | **70.2±0.9** |
> | Qwen3-8B-V2 (LoRA)| 61.4±3.0 |
> ||
> | Llama3.1-8B | 49.1±2.48 |
> | Llama3.1-8B-V2 | **66.69±3.8** |
>
> WONDA yields substantial gains on Qwen3-0.6B (22.8%→53.5%) and newly trained Llama3.1-8B-Instruct (49.1%→66.7%), yields no significant change on Qwen3-4B, and regresses on Qwen3-8B (70.2%→61.4%). Combined with the easy-split results (Table T1), these experiments provide evidence that WONDA's training signal transfers beyond our evaluation set. We do agree that studying the conditions, many of which independent of the WONDA framework (e.g., training method) that reliably ensures OOD generalization would be an important next step. We will discuss this point in the paper.
>
> **W6.** We acknowledge the density of definitions and tried out best to make the presentation clear. We will work to improve the clarity in the revised version. If there are specific parts that the reviewer found unclear, we would appreciate you pointing it out.
>
> We hope the additional experiments and analysis help address your concerns.

---

> > ### Author Rebuttal · Reviewer_76Vk · 2026-04-03
> >
> > Thank you for the detailed response! Really appreciate all the experiments you did in a short time frame. The plots and the figures really help understand the rebuttal. But, I still have some concerns:
> >
> > 1. W4 - I would have loved to see some experiments demonstrating iterative refinement of the invariants specifically for the hard case. Lemur cites this work : A New Era in Software Security: Towards Self-Healing Software via Large Language Models and Formal Verification [1] which was using counterexamples to refine the program (not exactly invariant refinement). The experiment need not be complex: a simple boolean feedback signal ("this invariant is incorrect, condition X is failing") up to directly including the SMT counterexample in the prompt would suffice as a baseline. The natural language interface of the LLMs makes this easy to prototype.
> >
> > [1] A New Era in Software Security: Towards Self-Healing Software via Large Language Models and Formal Verification. Tinhanyi et al.
> >
> > 2.W5 - For larger models (2/4), fine-tuning on WONDA data hurts correctness on Code2Inv compared to the base model. This is slightly concerning, which is related to whether WONDA is actually helping models become better at reasoning about invariant generation.
> >
> > 3. W1 Fig R2 is helpful and it would be great if a discussion could be included in the main paper. However, correctness gap is around 72% with gpt-5.2. But the inference overhead is also improving for these frontier models. Invariant generation competition like SyGuS-INV have correctness as the primary criterion. The alternative to automated invariant generation is manual verification, which is extremely time-consuming, even a system that takes an hour to generate a correct proof provides enormous practical value. I feel this makes correctness the more durable metric to optimize for in the long run.

---

> > > ### Author Response · Authors · 2026-04-05
> > >
> > > We thank the reviewer for the engaging discussion!
> > >
> > >
> > > > W4 - I would have loved to see some experiments demonstrating iterative refinement of the invariants specifically for the hard case. Lemur cites this work : A New Era in Software Security: Towards Self-Healing Software via Large Language Models and Formal Verification [1] which was using counterexamples to refine the program (not exactly invariant refinement).
> > >
> > > We agree that CEGAR-style refinement is an important direction for invariant generation. However, our current work focuses on improving an LLM’s zero-shot capability to autonomously infer invariants, which is a setting considered in prior work.
> > > Using LLMs to refine invariants based on counterexamples is unexplored. As the reviewer acknowledged, [1] focuses on program synthesis/repair. We will include a discussion of this direction in the revision.
> > >
> > > > W5 - For larger models (2/4), fine-tuning on WONDA data hurts correctness on Code2Inv compared to the base model. This is slightly concerning, which is related to whether WONDA is actually helping models become better at reasoning about invariant generation.
> > >
> > > OOD generalization is an unsolved problem and not something that should be automatically expected from the trained model. Currently, for a model to do well across a wide range of benchmarks (across different programming languages and program distributions), the best-known approach is to train the model on a variety of data. Our work focuses on labeling programs with high-quality invariants and relies on existing training programs (InvBench). For this reason, we find it encouraging that WONDA out-of-the-box can sometimes already generalize to OOD test data, including on one of the two largest models we considered (Llama-3.1-8B).
> > >
> > > > W1 Fig R2 is helpful and it would be great if a discussion could be included in the main paper.
> > >
> > > Thank you for the feedback! We will include and discuss Figure R2 in the main text.
> > >
> > > > However, correctness gap is around 72% with gpt-5.2.
> > >
> > > We clarify that GPT-5.2’s correctness rate is 72.4%, and that of Qwen3-4B improves from 22.8% to 44.4%.
> > >
> > > > But the inference overhead is also improving for these frontier models.
> > >
> > > Just as the inference efficiency for frontier models is improving, so is the capability of small models, which can amplify the effectiveness of our fine-tuning approach. Plus, the improved inference time might come at an increased cost and energy footprint. Therefore, achieving competitive end-to-end verification performance with small models is valuable.
> > >
> > > > Invariant generation competitions like SyGuS-INV have correctness as the primary criterion.
> > >
> > > We agree that correctness is a very important metric, which is why WONDA’s ability to significantly improve correctness is one of the contributions.
> > >
> > > > Even a system that takes an hour to generate a correct proof provides enormous practical value.
> > >
> > > We are focused on improving fully automated verifiers such as UAutomizer, where end-to-end verification performance is the key metric. We do agree that what the reviewer describes can be true in some other cases (e.g., for semi-automated tools such as Dafny and Verus).

---

### Official Review · Reviewer_aprm · 2026-03-13

**Soundness:** 3
**Presentation:** 4
**Significance:** 2
**Originality:** 2
**Overall Recommendation:** 4
**Confidence:** 3

**Summary:**

This work proposes WONDA, a pipeline for generating and curating high-quality training data for loop-invariant generation with (S)LMs for program verification. Concretely, WONDA consists of two steps: 1) AST-based normalization of verifier-generated invariants, and 2) semantic rewrites to improve clarity, generalization, and efficiency. Resulting invariants are scored based on correctness, sufficiency, and speedup. Retaining only invariants satisfying sufficiency and correctness leads to a training set of 7k high-quality invariants collected from 4k programs. Training Qwen3 family SLMs on these using SFT (with LoRA for the 8B model) yields significantly improved invariant quality and verification speedup on a dataset of hard verification instances, leading to ~2 of 20 previously timed-out instances being verified. Ablations show that in particular, the semantic rewrite is critical for fine-tuning success.

**Compliance With Llm Reviewing Policy:**

Affirmed.

**Final Justification:**

The reviewers have successfully addressed my concerns during the rebuttal. Their approach seems promising in a relevant field and experiments and ablations were conducted thoroughly. My main remaining concern is the focus on SLMs and relatively small-scale training data, limiting the overall performance compared to general-purpose frontier models. I believe these would need to be addressed to strongly recommend this work for acceptance. As is, it however already clears the bar and can provide a valuable foundation for future work in the area.

**Key Questions For Authors:**

- Can you compare to Wei et al. 2025's approach at least for one of the models in Table 3?
- Can you conduct full FT experiments on the 8B (and preferrably even larger) models?
- Can you report the total training set size (Appendix C seems to indicate roughly 3.7M?)
- Have you experimented with considering only hard or grade 3 samples from the training set? (possibly in a curriculm setting)

**Limitations:**

* Limitations are discussed although LLM inference costs are not represented adequately

**Strengths And Weaknesses:**

## Strenghts:
- The paper is very well motivated and timely, with an increasing accessibility of program verification
- WONDA is clearly effective as demonstrated in both the main results and ablation.
- The paper is very well written: Instructive running example, great introduction to notation and required background, well structured method and evaluation
- VBP is an interesting new metric, capturing the possible advantage of using LLM-generated invariants more realisticly. However, it ignored the ability to run two verification queries in parallel and does not consider the compute/time/cost of generating the invariants. While this is captured in VBP_E2E, the actual hardware required would correspond to a much longer timeout for the baseline.

## Weaknesses:
- Performance of WONDA FTed SLMs still falls noticeably short of larger open weight and frontier models and advantages over a FLOP matched solver are not reported (LLM inference is extremely computationally expensive). Baseline results (solver only) are generally missing in Table 3, making some improvements hard to assess.
- While WONDA is motivated with the Teacher Ceiling Trap, it does not argue how it can escape it, as it is ultimately distilling invariants generated by the same analyzer.
- Verifier-based RL would be a perfect fit for this problem and could escape said trap but is not considered in this work.
- Scaling studies are extremely limited with only 7k training samples (no token amount reported as far as I could find). The 8B model is only LoRA fine-tuned (not mentioned in the main work) leaving only two comparable results (4B and 0.6B) for only one of which ablations are conducted
- The intro mentions Wei et al also fine-tune LLMs for invariant generation, but their approach is not compared against in Table 3.

## Nits:
* When using citations as grammatical part of the sentence /citet should be used to remove parenthesis

---

> ### Author Rebuttal · Authors · 2026-03-31
>
> We thank the reviewer for the thorough and constructive feedback.
>
> **W1.** Thank you for these points. We acknowledge that there is still a gap between WONDA-trained SLMs and large open-weight and frontier models in terms of the quality of the generated invariant. However, as demonstrated in our experiments, in practice, if we take inference time into consideration, fine-tuned 4B model (165.7s) surpasses GPT-OSS-120B (167.6s) and nearly matches GPT-5.2 (163.4s), so small models can still be competitive on end-to-end verification time. Furthermore, as shown in our response to W4, Qwen3-14B-V2 outperforms the 5.7x larger Qwen3-Next-80B, demonstrating that WONDA can close the gap with larger models. In terms of FLOP-matched comparison to the solver: in our setting, SLM inference is only 1-3s wall-clock while solver runs average 193s , so total cost is solver-dominated; VBP and VBP_E2E are nearly identical across our model sizes for that reason, and larger LLMs would only widen the inference gap relative to SLMs. We will add solver-only mean VBP (**193s**, as in Appendix A, Fig. 4) explicitly to Table 3.
>
> **W2.** Thank you for raising this point. The "overfitting trap" occurs if a model is trained exclusively on the outputs of the symbolic solver, which we seek to improve. To address this issue, **our key insight is to create training data that neither the symbolic solver nor the language model could produce on its own**. To do so, we begin with raw, verbose, and often ungeneralized invariants produced by the symbolic solver. We then utilize a combination of analytic rules and LLM-based rewriting to transform these raw invariants into elegant, more insightful invariants. These rewritten versions are verified by the symbolic solvers. This process results in novel and mathematically sound invariants used to fine-tune the model, which empirically resulted in models that can propose insightful invariants for complex programs that would have remained out of reach for the symbolic solver on its own.
>
> **W3.** We agree that RL is a natural next step, as mentioned in the Conclusion section, for which WONDA serves as a necessary foundation: RL from a weak policy (23% correctness) risks reward hacking and sample inefficiency. WONDA's curated SFT (44% correctness) provides a strong initialization, and our graded quality signal (correctness, sufficiency, speedup) defines a natural reward structure for RL.
>
> **W4 / KQ2 / KQ3.** We will mention in the main text that the 8B model was trained using LoRA. We add a new ablation for Qwen3-4B full FT in [Table T3](https://tinyurl.com/47uhmfv9), confirming gains are robust. To address Q2, we performed full fine-tuning on two additional models. Hard-split evaluation results (single run) shown in [Table T4](https://tinyurl.com/46c94c56). These results directly address the concerns about SLM performance relative to larger models (W1) and KQ2: Qwen3-14B-V2 outperforms the 5.7x larger Qwen3-Next-80B-A3B-Instruct on both VBP (157.4s vs 171.3s) and speedup rate (26.0% vs 20.3%), while LLaMA 3.1-8B-V2 more than doubles correctness (19.5%→44.7%).
>
> **On the limited scaling studies.** Due to time and cost constraints, in the LLM simplification stage we choose N=4, which resulted in total of 7k training samples as mentioned, we believe that choosing higher N will result in larger training dataset which in turn will translate to better performance and is left for future work. We report also the total training size (total token count): 3,767,698 (7,284 samples, mean: 517.3, min: 376, max: 1,023); will be stated explicitly in the revision.
>
> **W5 / KQ1.** V0 models are our closest proxy to Wei et al.: both train on raw solver-generated invariants from the same InvBench program set. Exact replication was not possible as their hyperparameters and prompt template are not public. There is also a task-level difference: they jointly select loop location and generate the invariant, whereas we specify the location explicitly. Despite this, V0 should be interpreted as a reasonable proxy rather than an exact reproduction. The V0-to-V2 improvement at every scale in Table T3 and the ablation study demonstrates WONDA's benefit. We will make this connection explicit in the revision.
>
> **KQ4.**  Yes, we have experimented with only grade 3 training and empirically observed inferior performance compared to grade >= 2 set. Our intuition is that data volume (7,284 samples vs 2767 samples) plays a critical role. Curriculum training is a promising direction we will discuss in the revision.
>
> **Nits.** `\citep` to `\citet` will be fixed throughout.
>
> **Limitations.** We agree LLM inference cost should be discussed more explicitly relative to solver time in limitations.

---

> > ### Author Rebuttal · Reviewer_aprm · 2026-04-01
> >
> > I thank the authors for the detailed and thoughtful rebuttal.
> > A follow-up question: Does the reported wall clock time for LLM inference include the on-machine inference of the LLM or is some hosting solution used? On a CPU server typically used for verification, these inference times seem short even for SLMs.
> >
> > I believe this work (narrowly) meets the bar for acceptance and will provide a valuable foundation for follow-up work. To more clearly recommend this work for acceptance, I believe the scalability of the approach and performance improvements with more training data would need to be demonstrated.

---

> > > ### Author Response · Authors · 2026-04-02
> > >
> > > > Does the reported wall clock time for LLM inference include the on-machine inference of the LLM or is some hosting solution used? On a CPU server typically used for verification, these inference times seem short even for SLMs.
> > >
> > > The SLM inference runs on a GPU on the same machine (not CPU-only), which accounts for the short inference times. Specifically, all experiments (both baseline UAutomizer and LLM-assisted verification) were conducted on the same SLURM cluster node (8 AMD EPYC 9354 cores, 256 GB RAM, NVIDIA L40S GPU), as described in Appendix F, ensuring a fair apples-to-apples comparison. For the larger models, GPT-OSS-120B was accessed through the Together AI API and GPT-5.2 through the OpenAI API, but the verification queries were still executed on the same node to ensure consistent resource allocation across all experiments. We will make this connection more prominent in the main text alongside the VBP_E2E results.
> > >
> > > We expect WONDA’s performance to improve as additional training data becomes available. We believe the pipeline itself is not a bottleneck: it scales linearly with the number of programs.
> > >
> > > We thank the reviewer for the constructive engagement during the review process.

---

### Decision · Program_Chairs · 2026-04-30

**Decision:**

Accept (regular)

**Comment:**

The paper introduces WONDA, a two-stage pipeline designed to curate high-quality training data for loop-invariant generation. The pipeline addresses the unidiomatic and verbose nature of invariants produced by symbolic solvers like UAutomizer. It first uses AST-based normalization to clean the symbolic output and in the second stage, it uses an LLM to perform semantic rewrites to improve clarity and generalization while maintaining correctness. The curated data is used to finetune the Qwen3 family as the authors evaluate performance using VBP (Verification-Based Performance), a metric capturing both the correctness of the invariant and the speedup it provides to the underlying verifier.

The reviewers agree that the general methodology is intuitive and interesting, and the ablation studies shows the effectiveness of each stage. Also it is appreciated that this paper aims to improve smaller language models (e.g., 4B) and through experiments it demonstrates that it can match or even exceed the verification speed of much larger models like gpt-oss-120b. The authors provided a great deal of clarifications and extra experiment results to address the concerns of the reviewers in the rebuttal phase, and a number of them are properly addressed. I'd recommend the authors to incorporate the discussions and new results into the next version of the paper, including but not limited to: 1) discussion on the correctness vs. speed in terms of evaluating formal verification; 2) discussion of iterative refinement; 3) additional results on Llama-3.1 and even more base model results if possible.